# HKDC1 promotes tumor immune evasion in hepatocellular carcinoma by coupling cytoskeleton to STAT1 activation and PD-L1 expression

Yi Zhang[1,6], Mingjie Wang[1,2,6], Ling Ye[3,6], Shengqi Shen[2], Yuxi Zhang[1], Xiaoyu Qian[1], Tong Zhang[2], Mengqiu Yuan[3], Zijian Ye[1], Jin Cai[1], Xiang Meng[1], Shiqiao Qiu[1], Shengzhi Liu[1], Rui Liu[3], Weidong Jia[4], Xianzhu Yang [5] ✉, Huafeng Zhang [3] ✉, Xiuying Zhong [2] ✉ & Ping Gao [1,2] ✉

Immune checkpoint blockade (ICB) has shown considerable promise for treating various malignancies, but only a subset of cancer patients benefit from immune checkpoint inhibitor therapy because of immune evasion and immune-related adverse events (irAEs). The mechanisms underlying how tumor cells regulate immune cell response remain largely unknown. Here we show that hexokinase domain component 1 (HKDC1) promotes tumor immune evasion in a CD8+ T cell-dependent manner by activating STAT1/PD-L1 in tumor cells. Mechanistically, HKDC1 binds to and presents cytosolic STAT1 to IFNGR1 on the plasma membrane following IFNγ-stimulation by associating with cytoskeleton protein ACTA2, resulting in STAT1 phosphorylation and nuclear translocation. HKDC1 inhibition in combination with anti-PD-1/PD-L1 enhances in vivo T cell antitumor response in liver cancer models in male mice. Clinical sample analysis indicates a correlation among HKDC1 expression, STAT1 phosphorylation, and survival in patients with hepatocellular carcinoma treated with atezolizumab (anti-PD-L1). These findings reveal a role for HKDC1 in regulating immune evasion by coupling cytoskeleton with STAT1 activation, providing a potential combination strategy to enhance antitumor immune responses.

Cancer cells employ diverse strategies to evade immune detection and surmount immune response, which is collectively termed adaptive immune resistance (AIR)[1,2]. The first experimentally defined and therapeutically validated mechanism of AIR was selective induction of programmed cell death 1 ligand 1 (PD-L1) by interferon-γ (IFNγ) in the tumor[3–6]. Although current therapies that disrupt tumor immune evasion through ICB, including antibody-mediated blockade of PD-L1 binding to its cognate receptor, PD-1, have shown potent antitumor

[1]School of Medicine, South China University of Technology, Guangzhou, China. [2]Medical Research Institute, Guangdong Provincial People's Hospital, Guangdong Academy of Medical Sciences, Southern Medical University, Guangzhou, China. [3]The Chinese Academy of Sciences Key Laboratory of Innate Immunity and Chronic Disease, School of Basic Medical Sciences, Division of Life Science and Medicine, University of Science and Technology of China, Hefei, China. [4]Anhui Key Laboratory of Hepatopancreatobiliary Surgery, Department of General Surgery, Anhui Provincial Hospital, the First Affiliated Hospital of USTC, Division of Life Science and Medicine, University of Science and Technology of China, Hefei, China. [5]School of Biomedical Sciences and Engineering, South China University of Technology, Guangzhou International Campus, Guangzhou, China. [6]These authors contributed equally: Yi Zhang, Mingjie Wang, Ling Ye. ✉e-mail: yangxz@scut.edu.cn; hzhang22@ustc.edu.cn; zxywawj@ustc.edu.cn; pgao2@ustc.edu.cn

efficacy in 15–20% of patients, whereas immune evasion and therapeutic resistance lead to weaker clinical responses through largely unknown mechanisms in the majority of patients[7–9]. Combination therapies thus provide a potentially effective approach for overcoming PD-1/PD-L1 resistance and increasing response rates. Unfortunately, clinical trials of anti-PD-1/PD-L1 combined with other antitumor drugs, in the absence of a strong mechanistic rationale, have failed to show synergistic effects[10,11]. Thus, novel combination therapies that integrate anti-PD-1/PD-L1 ICB with efficacy prediction markers are urgently needed.

Metabolic reprogramming in cancer cells not only provides bioenergetic and biosynthetic precursors for sustaining tumorigenesis and cancer development, but also regulates antitumor immune response by depleting available nutrients from TME, promoting immune-suppressive metabolite secretion, and by modulating immune signal molecule expression[12–15]. Strategies for altering cell metabolism could serve as a potentially fruitful direction for cancer therapies[16]. However, many metabolic pathways that are activated in tumor cells are also required for antitumor immune cell activity, consequently presenting difficulties in blocking those pathways while still benefiting patients. Therefore, targets that enable the specific suppression or alteration of cancer metabolism could potentially confer an effective antitumor immune response. Interestingly, metabolic enzymes have been recently revealed to display moonlighting functions that directly regulate gene expression in cancer cells. For example, Pyruvate kinase M2 (PKM2) translocates into the nucleus in epidermal growth factor receptor (EGFR) activated glioblastoma cells, where it associates with β-Catenin to facilitate its recruitment to the CCND1 promoter, leading to tumor cell proliferation and brain tumor development[17]. However, since tumor development is a complex process involving numerous metabolic enzymes and regulatory molecules, it is still unknown whether and which other metabolic enzymes can also regulate tumor immunity, and if so, by what regulatory mechanisms.

Hexokinase (HK)-dependent phosphorylation of glucose is the rate-limiting step in glycolysis. Five HK isozymes (including HK1-4 and HKDC1) have been identified in mammalian cells, each with distinct subcellular localization, kinetics, substrate specificity, and physiological functions[18]. In macrophages, inhibition of HK1-dependent glycolysis suppresses both pro-IL-1β maturation and caspase-1 activation in response to LPS and ATP, suggesting that HK1-dependent glycolysis plays an important role in regulating NLRP3 inflammasome activation[19]. HK2 can act as a pattern recognition receptor to detect the release of N-acetylglucosamine and activate NLRP3 inflammasome formation in macrophages[20]. Aberrantly enhanced aerobic glycolysis in glioblastoma cells promotes HK2 dissociation from mitochondria to phosphorylate IκBα as a protein kinase. IκBα phosphorylation results in its degradation and NF-κB-activation-dependent PD-L1 expression tumor immune evasion[21]. Aberrant expression of HKDC1, a recently identified human hexokinase, is closely correlated with overall survival in cancer patients[22,23]. However, knockdown or knockout by RNAi or CRISPR shows that HKDC1 is not essential for cell viability in pooled screens (Supplementary Fig. 1c). Furthermore, HKDC1 exhibits low glucose-phosphorylating activity in hepatocytes[24], suggesting that HKDC1 might have "moonlighting" functions beyond its canonical hexokinase activity.

In this work, we use clinical HCC samples and a series of HCC preclinical models to demonstrate the role of HKDC1 in promoting tumor immune evasion by coupling STAT1 activation with the cellular cytoskeleton to enhance PD-L1 expression in HCC cells. These collective findings highlight HKDC1 as a potential target for combinational immunotherapies.

## Results

### HKDC1 expression associated with HCC progression and tumor-infiltrating CD8+ T cell exhaustion

To explore the potential role of glucose metabolism enzymes in modulating clinical response or resistance to ICB therapy, we first screened transcriptomic data from HCC cases in The Cancer Genome Atlas (TCGA, accessions: phs000178), GO30140 and IMbrave150 cohorts from The European Genome-phenome Archive (EGA, accessions: EGAD00001008130), for differentially expressed metabolic genes, which revealed that HKDC1 was expressed at substantially higher levels in clinical HCC lesion samples compared to that in matched non-cancerous tissues (Supplementary Fig. 1a). Western blot analysis confirmed the enhanced HKDC1 protein levels in HCC lesions compared to paired non-cancerous tissues (Supplementary Fig. 1b). Moreover, elevated HKDC1 expression was significantly negatively correlated with progression free survival in HCC patients treated with atezolizumab (Fig. 1a). Interestingly, publicly available CCLE RNAi and CRISPR gene dependency data indicated that HKDC1 expression had no obvious relationship with viability in HCC cell lines (Supplementary Fig. 1c). Using an HCC mouse model established by injecting YAP5SA plasmid into wild-type (WT) and HKDC1 knockout (KO) mice, we found that HKDC1 knockout significantly suppressed HCC incidence and development (Supplementary Fig. 1d, e). These data suggested that HKDC1 could play a role in promoting HCC progression potentially through the regulation of antitumor immunity.

To explore this possibility, we examined the composition of tumor-infiltrating immune cell populations in YAP5SA-induced HCC mouse model of WT and HKDC1 KO mice, and our flow-cytometric results showed that the percentage of tumor-infiltrating PD1+ CD8+ T cells was significantly lower in HKDC1 KO mice than that in the WT group (Fig. 1b). Further flow-cytometric results revealed that the proportion of exhausted CD8+ T cells (i.e., PD-1+ or LAG-3+) was significantly decreased and CD8+ T cell activity (reflected by the proportion of IFNγ+ or Granzyme B+ (GzmB+) cells) was increased in HCC-bearing HKDC1 KO mice compared to that in tumor-bearing WT mice (Fig. 1c, d). In order to investigate whether the decreased exhaustion and increased activity of tumor-infiltrating T cells in HCC-bearing HKDC1 KO mice is caused by the effects of HKDC1 depletion specifically in tumor cells or immune cells, we examined the expression levels of HKDC1 in tumor cells and other cells in the liver from YAP5SA induced HCC-bearing mice. qPCR results showed that HKDC1 was expressed at substantially higher levels in tumor cells compared to other cells (Supplementary Fig. 2a). Then, we used an HCC mouse model established by hepatic portal vein injection of murine liver cancer cells expressing shmHKDC1s or Non-targeted control (NTC) and quantified exhaustion and activity of tumor-infiltrating CD8+ T cells. We found that tumor growth was suppressed in mice inoculated with mHKDC1 knockdown (KD) Hepa1-6 cells (Supplementary Fig. 2b). Furthermore, the infiltrated CD8+ T cells exhibited lower PD-1 and LAG-3 expression and higher IFNγ and GzmB expression in mice inoculated with mHKDC1 KD Hepa1-6 cells compared with those in NTC-inoculated mice (Supplementary Fig. 2c, d). Similar results were observed in Hep55.1c orthotopic tumor-bearing mouse models (Supplementary Fig. 2e–g). These data indicate that loss of HKDC1 in cancer cells enhanced the antitumor activity of tumor-infiltrating CD8+ T cells.

To investigate whether the antitumor effects of HKDC1 depletion were dependent on CD8+ T cells, a CD8 monoclonal neutralizing antibody (αCD8) was used to treat immune-competent mice subcutaneously inoculated with mHKDC1 KD or NTC-expressing Hepa1-6 cells. Tumor growth was significantly attenuated under mHKDC1 KD, but could be restored by αCD8 treatment (Fig. 1e). Flow-cytometric analysis verified the depletion of CD8+ T cells in mice treated with

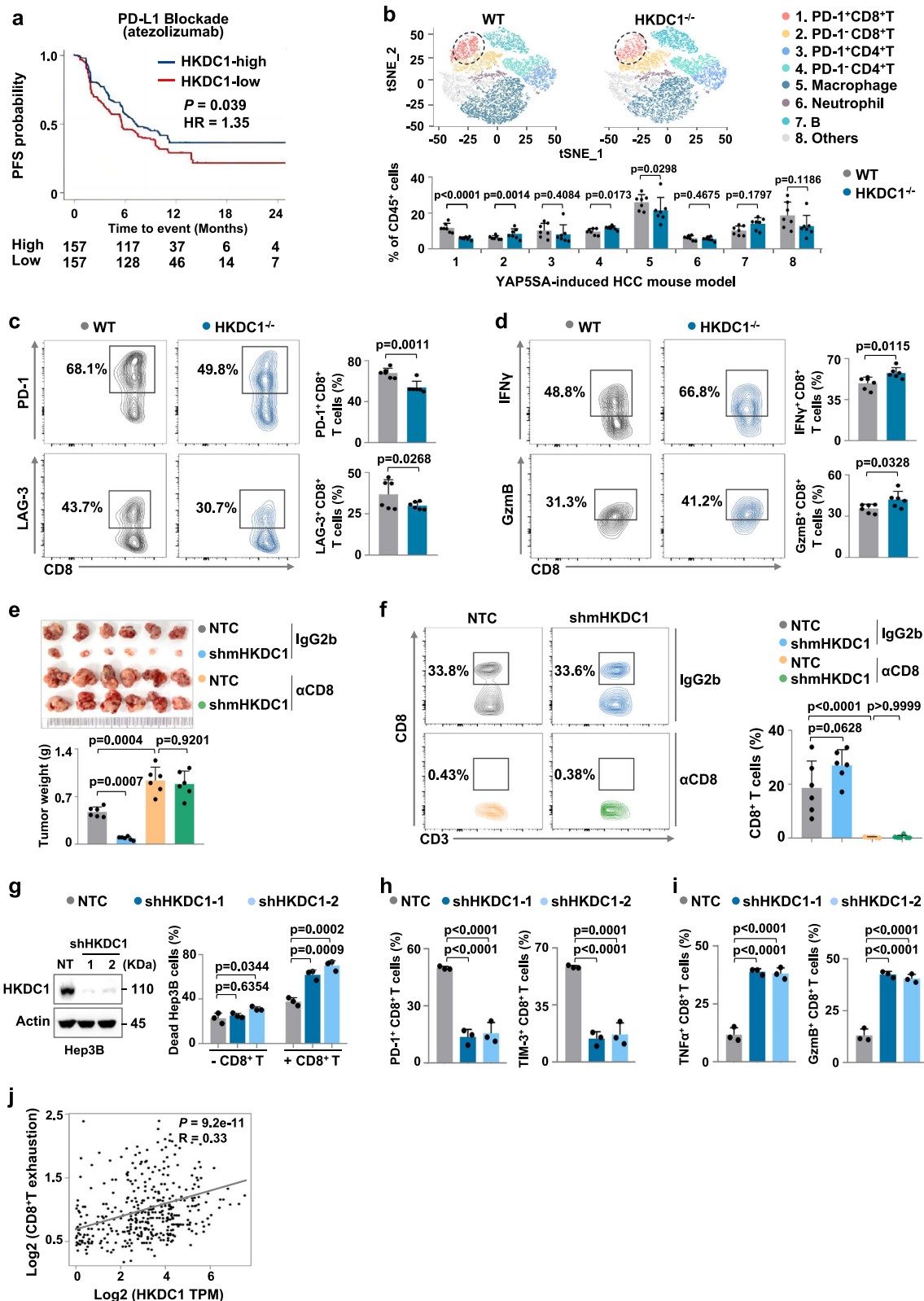

αCD8 (Fig. 1f). Through CD8+ T cell killing assays, we found that HKDC1 KD could sensitize Hep3B cells to cytolysis by CD8+ T cells (Fig. 1g), while flow-cytometric analysis showed that CD8+ T cells co-cultured with HKDC1 KD Hep3B cells displayed lower PD-1 and LAG-3, but higher IFNγ and GzmB expression (Fig. 1h, i), indicating tumor cell-mediated immune suppression was alleviated by HKDC1 depletion. Similar results of cell killing assays and flow-cytometric analysis were observed in mHKDC1 KD or NTC Hepa1-6 murine liver cancer cells and co-cultured CD8+ T cells (Supplementary Fig. 2h, i). In addition, transcriptomic data from clinical HCC samples in TCGA indicated that HKDC1 expression level was positively correlated with CD8+ T cell exhaustion (Fig. 1j). Collectively, these data indicated that aberrant HKDC1 expression in cancer cells promotes tumor immune evasion by increasing exhaustion of tumor-infiltrating CD8+ T cells.

**Fig. 1 | HKDC1 expression associated with HCC progression and tumor-infiltrating CD8+ T cell exhaustion. a** Kaplan-Meier curves for progression-free survival (PFS) of patients treated with Atezolizumab with low versus high expression of HKDC1. **b** The T-distributed stochastic neighbor embedding (t-SNE) plot showing the projection of various immune cells in livers from the YAP5SA-induced HCC mouse model of WT and HKDC1$^{-/-}$ mice ($n = 7$ male mice per group) is shown above, and the percentage of compared immune cells is shown below. **c** The percentage of tumor-infiltrating PD-1$^+$ or LAG-3$^+$ CD8$^+$ T cells in YAP5SA-induced HCC mouse model of WT and HKDC1$^{-/-}$ mice was analyzed by flow cytometry ($n = 6$ male mice per group). **d** The percentage of tumor-infiltrating IFNγ$^+$ or Granzyme B$^+$ (GzmB$^+$) CD8$^+$ T cells in YAP5SA-induced HCC mouse model of WT and HKDC1$^{-/-}$ mice was analyzed by flow cytometry ($n = 6$ male mice per group). **e** Tumor weight of Hepa1-6 xenografts expressing Non-targeted control (NTC) or shmHKDC1 with indicated treatment. Indicated Hepa1-6 cells were injected subcutaneously into C57BL/6 mice (6–8 weeks) with the treatment of 150 μg anti-CD8 antibodies (αCD8) or IgG2b isotype control twice a week ($n = 6$ male mice per group). Image of tumors was shown on the top. **f** The percentage of tumor-infiltrating CD8$^+$ T cells in the Hepa1-6 xenografts in **e** was analyzed by flow cytometry. **g** Cell death analysis of indicated Hep3B cells co-cultured with activated CD8$^+$ T cells separated from human PBMCs by flow cytometry. Western blot analysis of the protein levels of HKDC1 in indicated Hep3B. β-Actin (Actin) served as the loading control. **h** The percentage of PD-1$^+$ or TIM-3$^+$ CD8$^+$ T cells co-cultured with indicated Hep3B was analyzed by flow cytometry. **i** The percentage of TNFα$^+$ or GzmB$^+$ CD8$^+$ T cells co-cultured with indicated Hep3B cells was analyzed by flow cytometry. **j** Correlation between HKDC1 and CD8$^+$ T cell exhaustion in LIHC-tumor from TCGA dataset ($n = 369$ tumor samples). Data are presented as mean ± SD **b**–**f**. Data are presented as mean ± sem of three biologically independent experiments **g**–**i**. $P$-values were calculated by log-rank (Mantel-Cox) test **a**, two-tailed unpaired Student's t-test **b**–**d** or one-way ANOVA **e**–**i**. $P$-values and R were calculated by two-tailed Person's correlation analysis **j**. Source data are provided as a Source Data file.

## HKDC1 enhances PD-L1-mediated immune evasion of HCC cells independent of its hexokinase activity

To investigate whether HKDC1 hexokinase function in glycolysis plays a role in its promotion of tumor-infiltrating CD8$^+$ T cell exhaustion, we used AlphaFold2 to predict the HKDC1 protein structure and identify possible catalytic sites. Molecular docking prediction of mHKDC1 with glucose and ATP substrates highlighted residues S155, S600 and D654 as likely catalytic sites, but both glucose and ATP only interacted close to S600 (Fig. 2a). Subsequent in vitro hexokinase activity assays using WT mHKDC1 (mHKDC1$^{WT}$) and a series of likely catalytic site mutants (mHKDC1$^{S155A}$, mHKDC1$^{S600A}$ and mHKDC1$^{D654A}$) purified from *E. coli* showed that hexokinase activity was abolished upon disruption of S600 (Fig. 2b). In an HCC mouse model established by hepatic portal vein injection of mHKDC1 KD (shmHKDC1 targeting 3′-UTR of mHKDC1 transcripts) Hepa1-6 cells in immune-competent C57BL/6 mice, we found that overexpression of either mHKDC1$^{WT}$ or mHKDC1$^{S600A}$ could largely restore Hepa1-6 tumor growth suppressed by shmHKDC1, suggesting the role of HKDC1 in promoting tumor progression independent of its hexokinase activity in immune-competent mice (Fig. 2c and Supplementary Fig. 3a). Similar results were observed in xenograft model mice generated by subcutaneous inoculation with mHKDC1 KD or NTC Hepa1-6 cells overexpressing mHKDC1$^{WT}$ or mHKDC1$^{S600A}$ (Supplementary Fig. 3b, c). Further flow-cytometric analysis of tumor-infiltrating CD8$^+$ T cell exhaustion and activity markers revealed that overexpression of either mHKDC1$^{WT}$ or mHKDC1$^{S600A}$ could rescue PD-1 and LAG-3 expression suppressed by mHKDC1 KD (Fig. 2d), while decreasing the expression of activity markers (Fig. 2e). Similarly, mHKDC1$^{WT}$ or mHKDC1$^{S600A}$ expression could attenuate CD8$^+$ T cell-mediated tumor cell killing in vitro (Supplementary Fig. 3d–f). These cumulative data suggested that HKDC1 expression could enhance tumor immune evasion independently of its hexokinase function.

Amino acid alignments of HKDC1 protein sequences revealed that human HKDC1 contains a serine residue at position S602 corresponding to S600 in mHKDC1 (Supplementary Fig. 3g), and our CD8$^+$ T cell killing assays demonstrated that overexpression of either HKDC1$^{WT}$ or HKDC1$^{S602A}$ in HKDC1 KD Hep3B cells could restore resistance to CD8$^+$ T cell cytolytic activity (Fig. 2f). In addition, CD8$^+$ T cells co-cultured with HKDC1 KD Hep3B cells had lower levels of exhaustion and increased antitumor activity, which could be reversed by overexpression of either HKDC1$^{WT}$ or HKDC1$^{S602A}$ (Fig. 2g, h), suggesting that hexokinase activity was not required for the pro-tumorigenic effects of human HKDC1.

To explore how HKDC1 regulates CD8$^+$ T cell-dependent tumor immune evasion, we detected the mRNA levels of immune checkpoint markers in IFNγ-stimulated Hep3B cells expressing shHKDC1s or NTC. Relative expression analysis by qPCR indicated that HKDC1 KD significantly reduced the *CD274* (encoding PD-L1) expression elevated by IFNγ stimulation in Hep3B cells (Fig. 2i). Consistent with this result,

*Cd274* transcription was increased in the tumor tissues of YAP5SA-induced HCC mouse model, which was significantly suppressed by HKDC1 KO (Fig. 2j). Western blot analysis also showed that HKDC1 KD could obviously decrease PD-L1 protein levels accumulated by IFNγ stimulation of HCC cells compared to that in NTC cells (Fig. 2k and Supplementary Fig. 3h). Decreased PD-L1 protein levels were also observed in YAP5SA-induced HCC tumor tissues of HKDC1 KO mice compared with the WT group (Fig. 2l). These data suggested that HKDC1 could inhibit the antitumor effects of CD8$^+$ T cells by promoting PD-L1 expression in HCC cells. Supporting this possibility, HKDC1$^{WT}$ or HKDC1$^{S602A}$ overexpression restore the reduced PD-L1 expression resulting from HKDC1 KD in IFNγ-stimulated Hep3B cells (Fig. 2m). Similar results were observed in HepG2 cells (Supplementary Fig. 3i). Indeed, qPCR analysis of *Cd274* expression in tumor tissues from HCC model mice used in Fig. 2C showed that mHKDC1 KD significantly reduced *Cd274* transcription, while either mHKDC1$^{WT}$ or mHKDC1$^{S600A}$ overexpression resulted in similar *Cd274* expression to that in the NTC controls (Supplementary Fig. 3j). Moreover, using an HCC mouse model established by hepatic portal vein injection of Hepa1-6 murine liver cancer cells with indicated phenotypes, we found that tumor growth was promoted in mice inoculated murine liver cancer cells with mHKDC1 overexpressing (mHKDC1), but mPD-L1 knockout (sgmPD-L1) attenuated tumor growth promoted by HKDC1 overexpression (Supplementary Fig. 3k). Similar results were observed in Hep55.1c orthotopic tumor-bearing mouse models (Supplementary Fig. 3l). These results indicated that HKDC1 obviously promotes HCC growth through up-regulating PD-L1. Furthermore, qPCR analysis revealed a strong correlation between *HKDC1* and *CD274* expression in clinical HCC tissues (Fig. 2n). These results indicated that HKDC1 contributed to PD-L1-mediated immune escape in HCC cells in a manner independent of its hexokinase activity.

## HKDC1 associates with STAT1 and facilitates its phosphorylation to enhance PD-L1 expression

To explore the mechanism by which HKDC1 increases PD-L1 expression, we conducted immunoprecipitation (IP) assays with antibody targeting Flag-tagged HKDC1 in Hep3B cells followed by MS identification of candidate interaction partners. In particular, STAT1 that was predicted as transcription factor of CD274 was detected in HKDC1-interacting proteins (Fig. 3a). Subsequent IP assays in HCC cells confirmed that HKDC1 could associate with STAT1 (Fig. 3b and Supplementary Fig. 4a, b). Pull-down assays using glutathione S-transferase (GST)-fused HKDC1$^{WT}$ or GST-HKDC1$^{S602A}$ and His-fused STAT1 expressed and purified from *E. coli* showed that either form of HKDC1 could directly bind to STAT1, and further indicated that the S602 active site was not involved in HKDC1-STAT1 interactions (Fig. 3c). To investigate whether STAT1 participated in HKDC1-mediated promotion of PD-L1 expression, we detected *CD274* mRNA and PD-L1 protein levels in

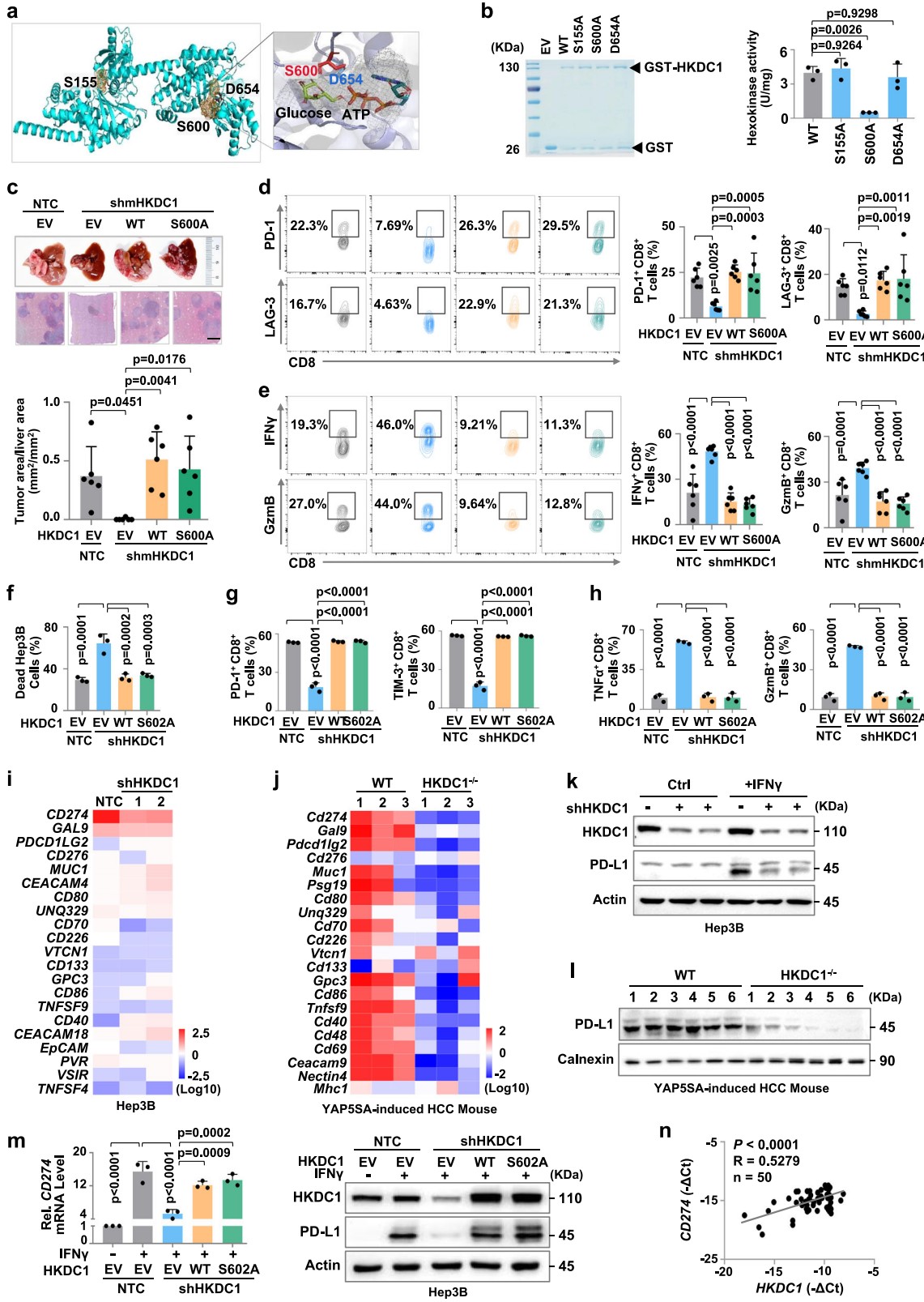

STAT1 KD HCC cells overexpressing HKDC1 with or without IFNγ stimulation. We found that HKDC1 overexpression resulted in elevated PD-L1 expression at both the mRNA and protein levels in HCC cells stimulated with IFNγ, and that this effect could be abolished by STAT1 knockdown (Fig. 3d and Supplementary Fig. 4c). These data indicated that STAT1 was indeed required for HKDC1-mediated transcriptional upregulation of PD-L1.

Western blot analysis showed that neither HKDC1 KD nor overexpression affected STAT1 protein levels in HCC cells (Supplementary Fig. 4d, e). However, WB analysis of STAT1 in the nuclear fraction of HKDC1 KD Hep3B cells and confocal fluorescence microscopy with antibody against STAT1 showed that STAT1 nuclear translocation was strikingly reduced by HKDC1 depletion (Fig. 3e, f). Since phosphorylation of STAT1 residue Y701 is essential for its nuclear translocation

**Fig. 2 | HKDC1 enhances PD-L1-mediated immune evasion of HCC cells independent of its hexokinase activity. a** Molecular docking model of mHKDC1 with glucose and ATP substrates. **b** in vitro hexokinase enzyme activity assay using GST tagged wildtype HKDC1 (GST-HKDC1[WT]) and HKDC1 mutants (GST-HKDC1[S155A], GST-HKDC1[S600A], GST-HKDC1[S654A]) purified from *E. coli*. **c** The H.E. quantification of tumor area/liver area (mm²/mm²) in hepatic portal mouse model. Indicated Hepa1-6 cells were injected through hepatic portal vein into C57BL/6 mice (6-8 weeks, *n* = 6 male mice per group). Images of liver and H.E. staining were shown on the top. Scale bars, 500 μm. **d** The percentage of tumor-infiltrating PD-1⁺ or LAG-3⁺ CD8⁺ T cells in hepatic portal mouse model in **c**. **e** The percentage of tumor-infiltrating IFNγ⁺ or GzmB⁺ CD8⁺ T cells in hepatic portal mouse model in **c**. **f** Cell death analysis of indicated Hep3B cells co-cultured with activated CD8⁺ T cells by flow cytometry. **g** The percentage of PD-1⁺ or TIM-3⁺ CD8⁺ T cells co-cultured with indicated Hep3B cells. **h** The percentage of TNFα⁺ or GzmB⁺ CD8⁺ T cells co-cultured with indicated Hep3B. **i** qPCR analysis of immune checkpoint markers in indicated Hep3B without

IFNγ stimulation. **j** qPCR analysis of immune checkpoint markers in tumor tissues of YAP5SA-induced HCC mouse model with indicated genotypes (*n* = 3 male mice per group). **k** Western blot analysis of PD-L1 in indicated Hep3B with or without IFNγ stimulation. **l** Western blot analysis of PD-L1 in tumor tissues of YAP5SA-induced HCC mouse model with indicated genotypes (*n* = 6 male mice per group). **m** qPCR analysis and Western blot analysis of PD-L1 in indicated Hep3B with or without IFNγ stimulation. Flag coding sequences were cloned on the C-terminus of HKDC1. **n** Correlation between the mRNA levels of *HKDC1* and *CD274* in clinical HCC tumor tissues was analyzed by qPCR (*n* = 50). Representative blot shown from three biologically independent experiments **k**, **m**. Data are presented as mean ± SD **c-e**, **j**. Data are presented as mean ± sem of three biologically independent experiments **b**, **f-i**, **m**. *P*-values were calculated by one-way ANOVA **b-h**, **m**. *P*-values and R were calculated by two-tailed Person's correlation analysis **n**. Source data are provided as a Source Data file.

and transcriptional regulatory activity[25], we performed WB analysis to examine its phosphorylation levels under HKDC1 KD or loss of hexokinase activity (i.e., in the HKDC1[S602A] variant). We found that STAT1-Y701 phosphorylation was obviously decreased in IFNγ-stimulated HCC cells with HKDC1 knockdown compared to that in stimulated NTC controls (Fig. 3g and Supplementary Fig. 4f), as well as in YAP5SA-induced HCC-bearing HKDC1 KO mice relative to tumor-bearing WT mice (Supplementary Fig. 4g). However, expression of either HKDC1[WT] or the hexokinase-deficient HKDC1[S602A] variant could restore the decreased STAT1 phosphorylation levels induced by HKDC1 knockdown (Fig. 3h and Supplementary Fig. 4h), suggesting that HKDC1 hexokinase activity was not involved in its effects on STAT1 phosphorylation. In addition, we induced STAT1 KD in HCC cells by expression of shSTAT1 (targeting its 5′UTR), then expressed STAT1[WT] or a STAT1[Y701A] mutant with blocked phosphorylation. Western blot analysis indicated that re-expression of STAT1[WT], but not STAT1[Y701A], could restore the STAT1 KD-induced decrease in PD-L1 protein levels in HCC cells, while concurrent overexpression of HKDC1 could further increase PD-L1 accumulation in cells overexpressing STAT1[WT], but not STAT1[Y701A] (Fig. 3i and Supplementary Fig. 4i). Notably, IHC analysis of 50 clinical HCC tumor samples showed that HKDC1 expression levels were positively correlated with STAT1 phosphorylation and PD-L1 protein levels (Fig. 3j). These results thus demonstrated that HKDC1 promotes STAT1 phosphorylation and nuclear translocation, together enhancing PD-L1 expression in cancer cells.

### HKDC1 presents cytosolic STAT1 to IFNGR1 to facilitate its phosphorylation via actin cytoskeleton protein ACTA2

We then investigated the possible mechanisms by which HKDC1 promotes STAT1-Y701 phosphorylation. Immunoprecipitation assays showed STAT1 interacts with IFNγ receptor 1 (IFNGR1) were obviously attenuated in HKDC1 KD Hep3B cells stimulated with IFNγ (Fig. 4a). Further Co-IP assays in Hep3B cells expressing HA-tagged HKDC1 and Flag-tagged IFNGR1 revealed that the HKDC1 and STAT1 could strongly interact with IFNGR1 under IFNγ stimulation (Fig. 4b). These results indicated that HKDC1 was necessary for STAT1 recruitment to IFNGR1.

Cytosolic STAT1 is known to be recruited to IFNGRs on the plasma membrane where residue Y701 is phosphorylated by JAK1/2 following exposure to IFNγ, and then phosphorylated STAT1 is transported into the nucleus to initiate a downstream transcriptional response[26,27]. However, the mechanism by which STAT1 in the cytosol is localized to IFNGRs on the plasma membrane under IFNγ stimulation is largely unclear[28]. Screening our MS data for other possible HKDC1 interaction partners revealed that HKDC1 could potentially bind to cytoskeletal proteins reportedly involved in signal transduction. Co-IP assays using antibody targeting endogenous or Flag-tagged HKDC1 showed that it could indeed interact with ACTA2, but not TUBA4A, in HCC cells (Fig. 4c and Supplementary Fig. 5a, b). Confocal fluorescence microscopy with antibodies against HKDC1 also verified that HKDC1

interacted with ACTA2 or Actin filaments (Fig. 4d, e). Immunofluorescence microscopy with WB analysis indicated that ACTA2 knockdown resulted in substantially reduced phosphorylation and nuclear localization of STAT1, as well as lower PD-L1 expression levels in HCC cells under IFNγ stimulation (Supplementary Fig. 5c, d).

To further investigate whether the role of HKDC1 in STAT1 phosphorylation requires association with Actin filaments, we detected STAT1 phosphorylation and PD-L1 expression levels in ACTA2 KD HCC cells overexpressing HKDC1. Western blots indicated that both IFNγ-stimulated STAT1 phosphorylation and PD-L1 expression were obviously increased by HKDC1 overexpression, but diminished under ACTA2 KD (Fig. 4f and Supplementary Fig. 5e). Furthermore, HKDC1 and STAT1 association with IFNGR1 was attenuated in ACTA2 KD Hep3B cells (Fig. 4g), suggesting that interaction with ACTA2 was necessary for HKDC1-mediated promotion of STAT1 recruitment to IFNGR1. Protein-protein molecular docking prediction of HKDC1, STAT1, and ACTA2 proteins showed that HKDC1 could potentially form several hydrogen bonds with STAT1 and ACTA2, whereas no direct bonding was predicted between ACTA2 and STAT1 (Supplementary Fig. 5f). Further IP assays confirmed that interaction between STAT1 and ACTA2 was greatly attenuated under HKDC1 KD, while ACTA2 KD had no apparent effect on interaction between HKDC1 and STAT1 (Fig. 4h, i and Supplementary Fig. 5g, h). Taken together, these data demonstrated that HKDC1 enhanced STAT1 phosphorylation in a manner dependent on its association with Actin filaments.

We next investigated whether ACTA2 was also involved in HKDC1-mediated tumor immune escape. The CD8⁺ T cell killing assays showed that the increased resistance to CD8⁺ T cell-mediated cytolysis induced by HKDC1 overexpression could be reversed by ACTA2 KD in Hep3B cells (Fig. 4j). Furthermore, ACTA2 KD could also reverse the increased exhaustion levels and decreased antitumor activity of CD8⁺ T cells co-cultured with HKDC1 overexpressing Hep3B cells (Fig. 4k, l). Collectively, these results illustrated that HKDC1 association with Actin filament is necessary for its promotion of tumor immune escape.

### HKDC1 inhibition in combination with PD-1/PD-L1 blockade enhances T cell antitumor response in HCC model mice

To validate the effects of HKDC1 inhibition as a potential intervention strategy for HCC progression in vivo, we inoculated Hepa1-6 cells via the hepatic portal vein in C57BL/6 immune-competent mice and employed a PLGA-based vesicle-like nanoparticle to deliver siRNAs targeting HKDC1 (VNP[siHKDC1/MTO]; siHKDC1 hereafter) or deliver control siRNAs (VNP[siCrtl/MTO]; siCtrl hereafter) into tumor cells. As a result, HKDC1 inhibition and low dose PD-1 antibodies (αPD-1[low], 25 μg/mouse) each modestly inhibited tumor growth, while siHKDC1 further enhanced the antitumor effects of αPD-1 (Fig. 5a). qPCR analysis confirmed that siHKDC1 significantly suppressed *Cd274* expression (Fig. 5b). Survival analysis showed that survival was significantly longer in mice treated with both siHKDC1 and αPD-1[low] compared with that in

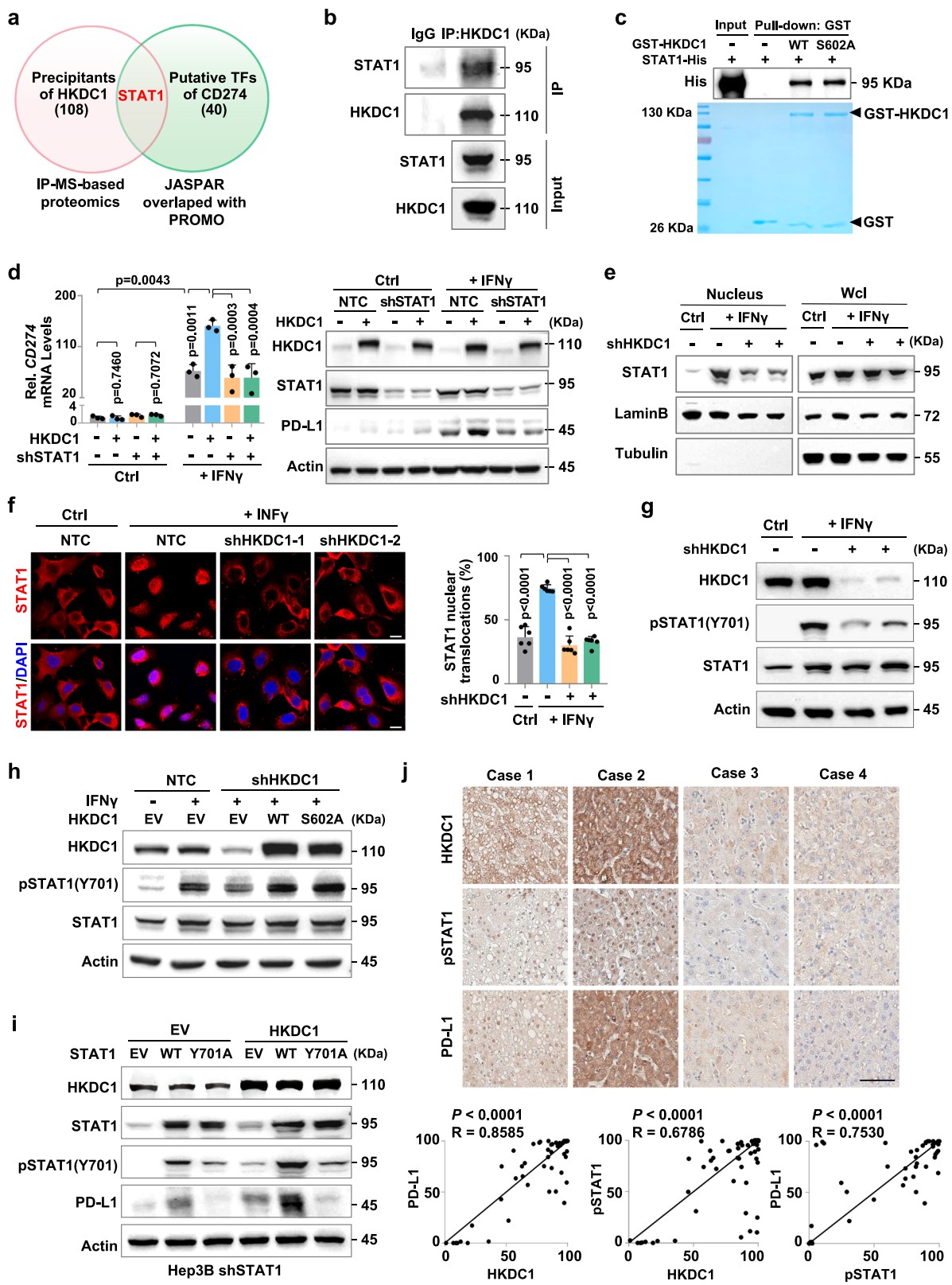

any other group (Fig. 5c). While ICB therapy holds promise for HCC patients, immune-related adverse reactions (irAEs), such as hyperglycemia, weight loss, and anemia, are common side effects of these treatments[29,30]. Notably, the combination treatment of siHKDC1 and αPD-1[low] exhibited similar antitumor effects with that of αPD-1[high] (250 μg/mouse) (Fig. 5a, b); however, median survival time was longer while the relative risk of irAEs was lower in the combination treatment

group compared to that in αPD-1[high] group mice (Fig. 5c and Supplementary Fig. 6a–e). Furthermore, flow-cytometric analysis indicated that the combination of siHKDC1 and αPD-1[low] resulted in significant suppression of exhaustion in tumor-infiltrating CD8[+] T cells, which displayed strikingly increased antitumor activity (Fig. 5d, e). Similar results were observed when combined siHKDC1 with αPD-L1 (Supplementary Fig. 6f–o).

**Fig. 3 | HKDC1 associates with STAT1 and facilitates its phosphorylation to enhance PD-L1 expression. a** Venn diagram showing the numbers of HKDC1-interacting proteins and predicted transcription factors of CD274. **b** Immunoprecipitation assay of interaction between endogenous HKDC1 and STAT1 in Hep3B cells. **c** Pull-down assay of the protein interaction between GST-HKDC1[WT] or GST-HKDC1[S602A] and STAT1-His. GST and 6×His coding sequences were cloned onto the N-terminus of HKDC1 and the C-terminus of STAT1, respectively. GST-tagged HKDC1[WT], GST-tagged HKDC1[S602A] and 6× His-tagged STAT1 proteins were purified from E. *coli* and incubated in vitro, followed by Western blot analysis with antibody against His-tag. **d** qPCR analysis and Western blot analysis of PD-L1 in indicated Hep3B cells with or without IFNγ stimulation. **e** Western blot analysis of STAT1 from nuclear fraction in indicated Hep3B with or without IFNγ stimulation. **f** Representative immunofluorescence images staining for STAT1 in indicated Hep3B with or without IFNγ stimulation. The nucleus was stained with DAPI. The percentage of nuclear STAT1 (right) in each group was quantitated using ImageJ.

Scale bars, 20 μm. **g** Western blot analysis of PD-L1, STAT1 and Tyr701 phosphorylated STAT1 in indicated Hep3B with or without IFNγ stimulation. **h** Western blot analysis of PD-L1, STAT1 and Tyr701 phosphorylated STAT1 in indicated Hep3B with or without IFNγ stimulation. **i** Western blot analysis of PD-L1, STAT1 and Tyr701 phosphorylated STAT1 in IFNγ-stimulated endogenous STAT1-knockdown Hep3B cells with indicated phenotype. **j** Representative IHC images of HKDC1, phosphorylated STAT1 (pSTAT1) and PD-L1 staining of clinical HCC tissues (*n* = 50 tumor samples). Scale bars, 100 μm. The correlation analysis of HKDC1, pSTAT1 and PD-L1 staining were shown below. Representative blot and immunofluorescence image shown from three biologically independent experiments **b**–**i**. Data are presented as mean ± SD **j**. Data are presented as mean ± sem of three biologically independent experiments **d** or six biologically independent experiments **f**. *P*-values were calculated by one-way ANOVA **d**, **f**. *P*-values and R were calculated by two-tailed Person's correlation analysis **j**. Source data are provided as a Source Data file.

We also validated the treatment effeciency of siHKDC1 combined with αPD-L1[low] in HCC xenograft model mice generated by subcutaneous inoculation of Hepa1-6 cells in C57BL/6 immune-competent mice. Consistent with our above model, in vivo tumor growth was significantly lower in mice treated with the siHKDC1-αPD-L1[low] combination compared to either siHKDC1 or αPD-L1[low] treatments alone (Supplementary Fig. 6p). The decrease in *Cd274* expression in tumors of siHKDC1-treated animals was confirmed by qPCR (Supplementary Fig. 6q). More importantly, the siHKDC1/αPD-L1[low] combined treatment also conferred significant survival benefits over that of either monotherapy (Supplementary Fig. 6r). Collectively, these results clearly demonstrated that HKDC1 inhibition enhances CD8+ T cell-mediated antitumor immunity by regulating PD-L1 expression and have the potential to become an important target for combinational immunotherapies.

## Discussion

The metabolic interplay between cancer cells and immune cells may be a determining factor in antitumor immune response[31,32]. Recent studies have provided evidence that targeting cancer and/ or immune cell metabolism can synergize with antitumor immunity[33–35]. Therefore, to better understand the role of metabolic crosstalk between tumor and immune cells, considerable research efforts have focused on the metabolic barriers to an effective antitumor immune response, such as metabolic competition in the TME, microenvironmental acidosis, immune-suppressive metabolite secretion, and metabolic reprogramming during immune cell development and activation. Several metabolic enzymes have been recently shown to harbor non-canonical ("moonlighting") functions that support tumor cell survival and proliferation[36–39]. However, the "moonlighting" functions of numerous metabolic enzymes and their regulatory effects on cancer immunity are still unknown. In this study, we uncovered an unexpected function of HKDC1 in regulating PD-L1-induced immune evasion independent of its hexokinase activity in glucose metabolism, thus providing insights into the role of metabolic enzymes in anti-cancer immunity. Our findings in HCC model mice also support the development of combination treatment strategies incorporating HKDC1 inhibition with ICB that could enhance antitumor effect while limiting the risk of irAEs.

IFNγ is a key cytokine largely produced by activated T cells to coordinate innate and adaptive antitumor immune response in the TME, while IFNγ signaling-mediated processes can ultimately induce feedback inhibition that promotes upregulation of inhibitory immune checkpoint molecules in tumor cells, leading to AIR[40,41]. Recent studies have highlighted the important role of STAT1 in PD-L1-induced tumor immune evasion under IFNγ stimulation[42,43]. However, STAT1 might not be a targetable factor, since it can act as either an oncoprotein or tumor suppressor in some malignant cells, depending on genetic background[44,45]. Intriguingly, although the JAK/STAT pathway is well-known to be activated under IFNγ stimulation, it is still unclear how cytosolic STAT1 is transported to IFNγ receptors on the membrane and phosphorylated by JAK to activate the JAK/STAT pathway[46]. Using IP followed by MS analysis along with confocal fluorescence microscopy, we observed that HKDC1 unexpectedly localizes to actin filaments and interacts with both cytoskeleton protein, ACTA2 and STAT1 in HCC cells. It has been reported that some cytoskeletal proteins not only maintain structural integrity, but also contribute to signal transduction[47]. We found that the interaction between HKDC1 and ACTA2 is necessary for STAT1 association with IFNGR on plasma membrane, STAT1-Y701 phosphorylation, and thus PD-L1 expression following stimulation with IFNγ. In addition, recent studies highlight the contribution of the interplay between metabolism and cytoskeleton in cancer progression. In transformed non-small cell lung cancer cells, F-actin bundling spatially sequesters the E3 ligase TRIM21, thus reducing its substrate PFKP degradation and enhancing glycolysis[48]. Cells can program their death by inducing in response to an imbalance of cystine. Glucose starvation suppresses SLC7A11[high] tumor growth by inducing extensive disulfide bonding of the actin cytoskeleton and disulfidptosis[49]. Our data establish a previously unappreciated mechanism by which HKDC1 couples the cytoskeleton to STAT1/PD-L1-mediated tumor immune evasion.

ICB has shown promising outcomes as a treatment for various malignancies, including HCC[50]. HCC patients treated with anti-PD-1 monotherapy or anti-PD-L1 combined with anti-vascular endothelial growth factor-A typically exhibit modest objective response rates (ORRs) of 15% and 27.3%, respectively; however, approximately 20% and 56.5% of patients experience grade 3 or higher treatment-induced irAEs[11,30] (IMBRAVE150). Therefore, identifying new therapeutic targets and combination strategies is essential for improving these relatively low response rates to immunotherapies and to reduce treatment toxicity. By analyzing tumor samples from 358 atezolizumab-treated HCC patients enrolled in the GO30140 phase 1b or IMbrave150 phase 3 trials or obtained from the TCGA database, we found that aberrantly high HKDC1 expression in tumor tissues could predict shorter survival of HCC patients treated with anti-PD-1 therapy. Mechanistically, HKDC1 enhances PD-L1 expression by coupling the activation of its transcriptional factor, STAT1, with the cytoskeleton. More importantly, our data from HCC model mice revealed that combining an HKDC1 siRNA nanotherapy with anti-PD-1/PD-L1 treatment could enhance antitumor response without exacerbating irAEs, after adjusting for confounding factors, such as survival time, body weight, HCT, HGB and blood glucose level. Taken together, this study uncovers a therapeutic vulnerability for HCC based on the role of HKDC1 in promoting tumor immune evasion through recruitment of cytosolic STAT1 to IFNGR1 on the plasma membrane, and suggests that a combination treatment strategy targeting both HKDC1 and ICB could be an effective route for developing of HCC interventions.

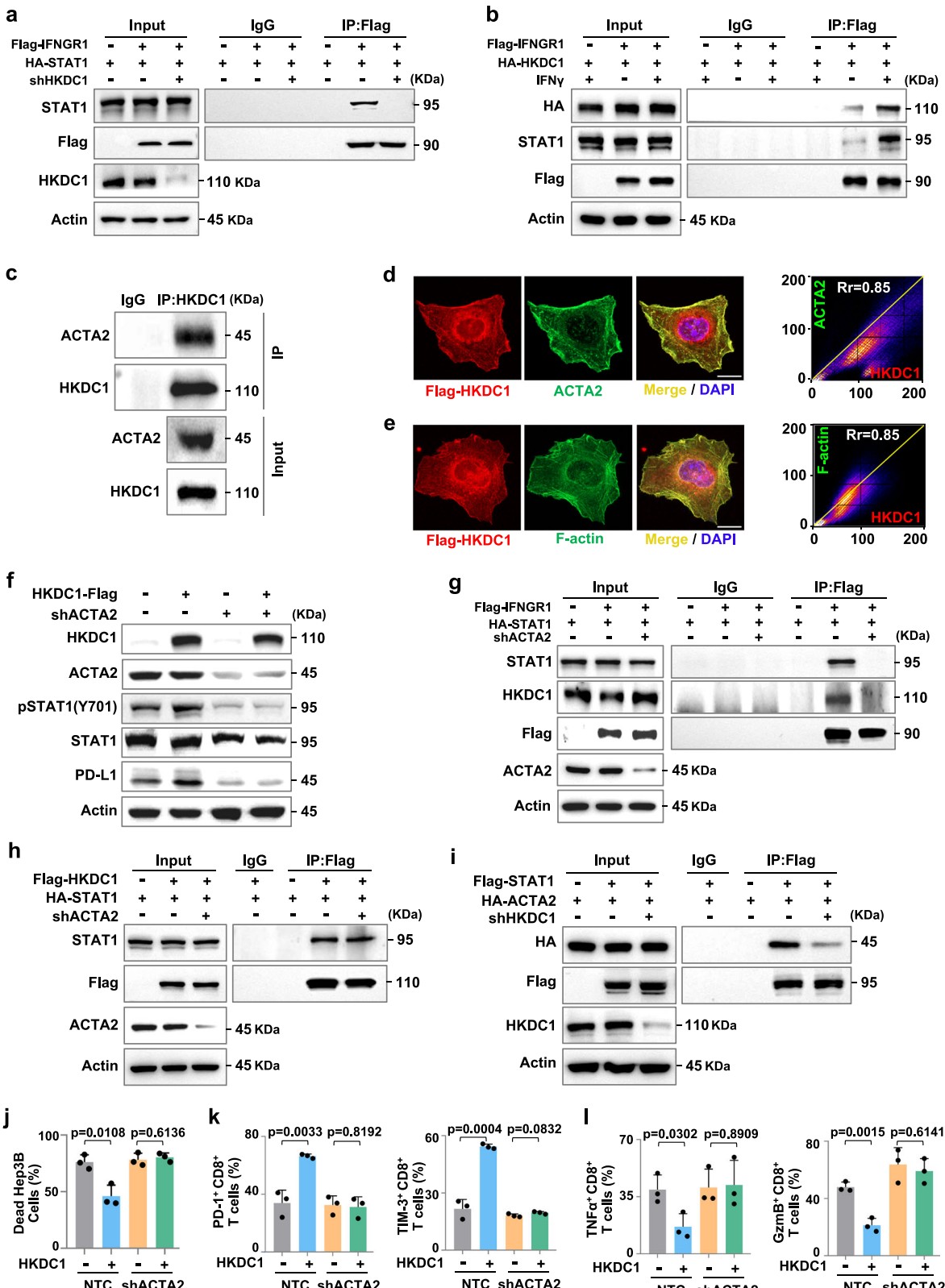

## Methods

Our research complies with all relevant ethical regulations of the South China University of Technology and the University of Science and Technology of China. All animal protocols were approved by the Animal Experiment Ethics Committee of the South China University of Technology (2023113) and were performed following the guidelines for the use of laboratory animals. The mice with orthotopic tumors,

authorized by the Committees on Animal Research and Ethics, consistently follow the humane endpoint. If the animal starts showing signs of immobility, a huddled posture, the inability to eat, ruffled fur, or self-mutilation, the animal will be euthanized immediately. The subcutaneous tumor maximum diameter was 20 mm and authorized by the Committees on Animal Research and Ethics and was not exceeded at any time during the experiments. The collection and use

**Fig. 4 | HKDC1 presents cytosolic STAT1 to IFNGR1 to facilitate its phosphorylation via actin cytoskeleton protein ACTA2. a** Immunoprecipitation assay of interaction between IFNGR1 and STAT1. Hep3B cells expressing NTC or shHKDC1 were transfected with HA-tagged STAT1 and Flag-tagged IFNGR1 plasmids and stimulated with IFNγ. **b** Immunoprecipitation assay of interaction between HKDC1 and IFNGR1. Hep3B cells were transfected with HA-tagged HKDC1 and Flag-tagged IFNGR1 plasmids with or without IFNγ stimulation. **c** Immunoprecipitation assay of interaction between endogenous HKDC1 and ACTA2 in Hep3B cells. **d** Representative immunofluorescence staining for Flag-tagged HKDC1 and ACTA2 in Hep3B cells. The nucleus was stained with DAPI. Co-localization analysis of immunofluorescence images were quantitated by the colocalization plugin. Scare bar, 20 μm. **e** Representative immunofluorescence staining for Flag-tagged HKDC1 and F-actin in Hep3B cells. The nucleus was stained with DAPI. Co-localization analysis of immunofluorescence images were quantitated by the colocalization plugin. Scare bar, 20 μm. **f** Western blot analysis of PD-L1, STAT1 and Tyr701 phosphorylated STAT1 in IFNγ-stimulated endogenous HKDC1-knockdown Hep3B cells with indicated genotypes. **g** Immunoprecipitation assay of interaction among HKDC1, STAT1, and IFNGR1. Hep3B cells expressing NTC or shACTA2 were transfected with HA-tagged STAT1 and Flag-tagged IFNGR1 plasmids and stimulated with IFNγ. **h** Immunoprecipitation assay of interaction between HKDC1 and STAT1. Hep3B cells expressing NTC or shACTA2 were transfected with HA-tagged STAT1 and Flag-tagged HKDC1 plasmids. **i** Immunoprecipitation assay of interaction between ACTA2 and STAT1. Hep3B cells expressing NTC or shHKDC1 were transfected with HA-tagged ACTA2 and Flag-tagged STAT1 plasmids. **j** Cell death analysis of indicated Hep3B cells co-cultured with activated CD8[+] T cells separated from human PBMCs by flow cytometry. **k** The percentages of PD-1[+] or TIM-3[+] CD8[+] T cells co-cultured with indicated Hep3B were analyzed by flow cytometry. **l** The percentages of TNFα[+] or GzmB[+] CD8[+] T cells co-cultured with indicated Hep3B cells were analyzed by flow cytometry. Representative blot and immunofluorescence image shown from three biologically independent experiments **a**–**i**. Data are presented as mean ± s.e.m of three biologically independent experiments **d**, **e**, **j**–**l**. *P*-values were calculated by two-tailed unpaired Student's *t*-test **j**–**l** or by the Pearson correlation test **d**, **e**. Source data are provided as a Source Data file.

of clinical materials were approved by the Institutional Research Ethics Committee of the First Affiliated Hospital of University of Science and Technology of China.

## Cell culture and reagents

Human Hep3B, HepG2, HEK293T, and mouse Hepa1-6 cell lines were purchased from the American Type Culture Collection. Mouse Hep55.1c cell line was purchased from the BioVector NTCC Inc. All cell lines were tested for mycoplasma contamination and no cell lines were contaminated. All cells were cultured in DMEM (Gibco) containing 10% fetal bovine serum (Gibco) and 1% penicillin-streptomycin and were kept in a humidified incubator at 37 °C and 5% $CO_2$. Cells were treated with 100 U/ml human IFNγ or 10 ng/ml murine IFNγ to induce STAT1 phosphorylation and PD-L1 expression. Cells were treated with 10 μmol/L Cytochalasin B to relax the microfilament.

## Plasmid construction and establishment of stable cell lines

shRNAs against HKDC1, STAT1 and ACTA2 were constructed into the pLKO.1 vector (Sigma-Aldrich). sgRNAs against PD-L1 were constructed into the lentiCRISPR v2 vector (lentiCRISPR v2 vector was gifted by Rongbin Zhou, University of Science and Technology of China). The target sequences of all shRNAs and sgRNAs used in this study are summarized in Supplementary Table 1. HKDC1, STAT1, ACTA2, or IFNGR1 were subcloned into the pSin-3×Flag, pSin-HA, or pMX-GFP empty vector; then they were cotransfected with plasmids encoding VSVG and Δ8.9 into HEK293T cells using PEI. Culture medium containing virus particles was collected 48 h post-transfection and added into the culture medium of Hep3B, HepG2, or Hepa1-6 cells with 8 μg/ml polybrene following the selection with 0.5–1 μg/ml puromycin.

## Western blot assay

Cells were lysed using RIPA buffer (50 mM Tris-HCl, pH 8.0, 150 mM NaCl, 5 mM EDTA, 0.1% SDS, 1% NP-40) supplemented with protease inhibitor cocktail. Equal amounts of cell lysates were boiled and fractionated by SDS-PAGE. Primary antibodies used for immunoblotting included anti- HKDC1, STAT1, STAT1 (Phospho-Tyr701), PD-L1, ACTA2, β-Actin, LaminB, α-Tubulin, Calnexin, HA-Tag, Flag-Tag, GFP-Tag, GST-Tag and His-Tag are summarized in Supplementary Table 3.

## qPCR analysis

According to the manufacturer's instructions, total RNA was extracted by TRIzol reagent, and 1–3 microgram of total RNA was reverse transcribed with HiScript II 1st Strand cDNA Synthesis Kit. qPCR was performed using SYBR Green Master Mix. All samples were normalized to 18 S RNA. Primer sequences are summarized in Supplementary Table 2.

## Immunoprecipitation assay

Cells were lysed with NP-40 IP buffer (20 mM Tris-HCl pH 7.5, 150 mM NaCl, 1.5 mM $MgCl_2$, 2 mM EDTA, 1% NP-40) supplemented with protease inhibitors for 1–2 h on ice, followed by centrifugation. The supernatants were incubated with indicated antibody overnight at 4 °C, then incubated with protein A/G-conjugated beads for 2 h at 4 °C. The beads were washed four times with 0.5% NP-40 buffer and boiled in SDS-PAGE loading buffer. Protein samples were analyzed by western blot.

## Pull-down assay

The cDNAs encoding HKDC1 or HKDC1 mutants were cloned into pGEX-4T-1 vector (GE Healthcare), and cDNA encoding STAT1 was cloned into pET-22b (+) vector (Novagen) by ClonExpressTM II One Step Cloning Kit. The proteins HKDC1, HKDC1 mutants and STAT1 were produced in E. *coli* (DE3). The proteins were induced with 0.5 mM IPTG for 18 h at 16 °C. Purified His-tagged proteins and GST-fused proteins were incubated in pull-down buffer (150 mM NaCl, 50 mM Tris pH 7.5, 0.1% NP-40, 5 mM dithiothreitol). After incubation, the beads were pelleted and washed with pulldown buffer. Protein samples were analyzed by western blot.

## In vivo depletion of CD8[+] T cells

To deplete CD8[+] T cells in vivo, mice were injected intraperitoneally with 150 μg of anti-CD8 antibody (αCD8) 3 days before tumor inoculation and twice weekly thereafter to ensure sustained depletion of CD8[+] T cell subset during the experimental period. On the contrary, groups of mice treated with IgG2b isotype control.

## CD8[+] T cell isolation and activation

Mouse CD8[+] T cells were separated from C57BL/6 mice spleen with CD8[+] T cell enrichment kit. CD8[+] T cells were stimulated and cultured with plate-bound anti-CD3 (2 μg/ml) and fully supplemented tissue culture medium (RPMI plus 10% FBS, 1 μg/ml anti-CD28, 25 mM HEPES, 5 μM mercaptoethanol, 1% penicillin-streptomycin, 10 μg/ml IL-2) for 3 days.

Human CD8[+] T cells were separated from PBMCs derived from healthy donors (in order to employ these clinical materials for research purposes, prior written informed consent was obtained from the donors, and approval was secured from the Institutional Research Ethics Committee of the First Affiliated Hospital of the University of Science and Technology of China) with CD8[+] T cell enrichment kit. CD8[+] T cells were cultured in RPMI-1640 medium and activated with Dynabeads™ Human T-Activator CD3/CD28 for 1 week according to the manufacturer's instructions.

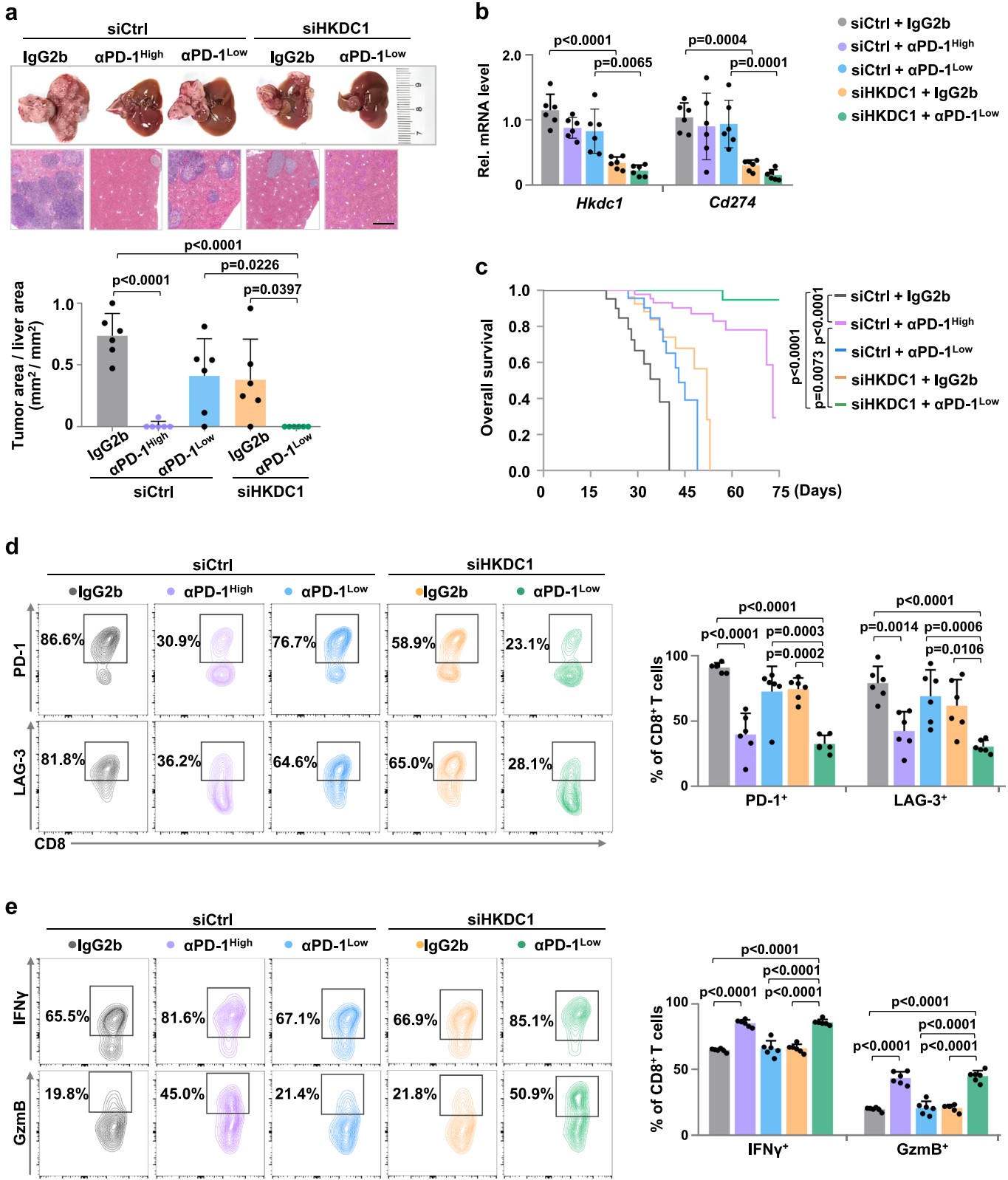

### T cell-mediated tumor cell killing assay

Hepa1-6 cells transfected with the indicated vector were pretreated with 10 ng/ml IFNγ for 6 h and co-cultured with activated mouse CD8$^+$ T cells at the ratio of 1:8 for 2 days, then harvested and labeled with Annexin V and PI. Stained Hepa1-6 cells were assessed by flow cytometry.

Hep3B cells stably transfected with indicated vector were pretreated with 100 U/ml IFNγ for 6 h and co-cultured with activated

human CD8$^+$ T cells at the ratio of 1:8 for 2 days, then harvested and labeled with Annexin V and PI. Stained Hep3B cells were assessed by flow cytometry.

### In vitro co-culture assay

Hepa1-6 cells stably transfected with indicated vector were pretreated with 10 ng/ml IFNγfor 6 h and co-cultured with activated mouse CD8$^+$

**Fig. 5 | HKDC1 inhibition in combination with PD-1 blockade enhances T cell antitumor response in HCC model mice. a** The H.E. quantification of tumor area/liver area ($mm^2/mm^2$) in hepatic portal mouse model with indicated genotypes ($n = 6$ male mice per group) is displayed below. Hepa1-6 cells were injected through hepatic portal vein into C57BL/6 mice (6–8 weeks), and randomly divided into five groups with indicated treatment. $VNP^{Ctrl/MTO}$ or $VNP^{siHKDC1/MTO}$ (0.34 OD/per mice) was injected through tail vein, then anti-PD-1 antibody ($\alpha$PD-1) or IgG2b isotype control was injected intraperitoneally twice a week. Images of liver and H.E. staining are shown above. Scale bars, 1 mm. **b** qPCR analysis of mRNA levels of *Hkdc1* and *Cd274* in the hepatic portal mouse model in **a**. **c** Overall survival of the hepatic portal mouse model in **a**. **d** The percentage of tumor-infiltrating PD-1+ or LAG-3+ CD8+ T cells in hepatic portal mouse model in **a** was analyzed by flow cytometry. **e** The percentage of tumor-infiltrating IFN$\gamma$+ or GzmB+ CD8+ T cells in hepatic portal mouse model in **a** was analyzed by flow cytometry. Data are presented as mean ± SD **a**–**e**. *P*-values were calculated by one-way ANOVA **a**, **b**, **d**, **e** or by log-rank (Mantel-Cox) test **c**. Source data are provided as a Source Data file.

T cells at the ratio of 3:1 for 3 days, then CD8+ T cells were harvested and stained to analyze the percentage of PD-1+, TIM-3+, TNF$\alpha$+ and GzmB+ CD8+ T cells by flow cytometry.

Hep3B cells stably transfected with indicated vector were pre-treated with 100 U/ml IFN$\gamma$ for 6 h and co-cultured with activated human CD8+ T cells at the ratio of 3:1 for 3 days, then CD8+ T cells were harvested and stained to analyze the percentage of PD-1+, TIM-3+, TNF$\alpha$+ and GzmB+ CD8+ T cells by flow cytometry.

### Flow cytometry
Single cell suspensions were prepared from cells in culture or tumors of HCC-bearing mice. For tumor samples from HCC-bearing mice, single cell suspension was obtained by rapid and gentle stripping, physical grinding at 4 °C or collagenase IV enzymatic hydrolysis for 1 h at 37 °C, and filter filtration. Tumor cells were collected using a tumor cell isolation kit, then other cells were suspended with 40% percoll reagent and centrifuged to obtain enriched immune single cells in cell precipitation and enriched CD45- non-tumor cells in the top layer. Enriched immune single cells were blocked with CD16/CD32 antibody and stained with indicated fluorochrome-conjugated antibodies for 30 min at 4 °C. To incubate the following antibodies: anti-IFN$\gamma$, anti-TNF$\alpha$, and anti-Granzyme B, cells should be permeabilized with a Cytofix/Cytoperm Kit. All data were collected using a BD Fortessa or Cytek NL-3000 and analyzed using FlowJo (v 10.8.1) software. Gating and sorting strategies were provided in Supplementary Fig. 8. The antibodies used in this study are listed in Supplementary Table 3.

### Immunofluorescence staining
Cells were fixed with Paraformaldehyde-glutaraldehyde at room temperature for 30 min. Then the membrane was broken with Triton X-100 for 30 min, followed by blocking with 5% BSA. The primary antibody was incubated at 4 °C overnight, and the secondary antibody was incubated at room temperature for 1 h. Finally, anti-fluorescence quenching sealing solution (including DAPI) (Beyotime) is used for sealing. Images of IF staining were captured using a Zeiss 710 laser confocal microscope (v Zen 10.0) and data were analyzed using ImageJ software (v 1.53k). Primary antibodies or reagent against the targeted proteins were used: Flag-Tag, STAT1, ACTA2, PhalloidiniFluor 488 and $\alpha$-Tubulin. Anti-rabbit or anti-mouse secondary antibodies conjugated to CoraLite488 or CoraLite594 were used.

### Histological analysis
Liver tissues were fixed with 10% neutral buffered formalin for 48 h, followed by dehydration and paraffin embedding. Sections (4 µm) were used for staining with hematoxylin and eosin. Whole-slide images were analyzed using Halo software (v 3.3.14).

### Clinical human HCC specimens and immunohistochemistry
Formalin-fixed, paraffin-embedded HCC tissues and adjacent non-cancerous tissues were collected from patients with HCC in the First Affiliated Hospital of University of Science and Technology of China. For the use of these clinical materials for research purposes, both previously obtained written informed consent from the patients and study approval by the institutional research ethics committee of the First Affiliated Hospital of the University of science and technology of China were obtained. As for IHC Assay, samples were deparaffinized in xylene, then rehydrated with graded ethanol. After antigens retrieval, incubate slide with 3% hydrogen peroxide for 10 min to block endogenous peroxidase activity at room temperature. Next, sections were preincubated in normal goat serum for 15 min to prevent nonspecific staining and visualized with DAB. Primary antibodies against the following proteins were used for IHC: HKDC1, PD-L1, and STAT1 (Phospho-Ser727). Images were acquired with NIS-Elements Viewer and Zeiss Zen lite. The staining results of the tissues were then quantitated by the Halo software (v 3.3.14).

### Animal experiments
All animals were housed at appropriate temperature (22–24 °C) and humidity (40–70%) under a 12/12 h light/dark cycle with unrestricted access to food and water for the period of the experiment. Six- to eight-week-old male mice were used for all animal experiments unless otherwise stated. HKDC1 KO C57BL/6 mice were generated using CRISPR genome editing (HKDC1 target sequence: CACAG ACGTGGTGAACCGCC), while the control group were wild type C57BL/6 mice (WT mice). For YAP5SA-induced HCC mouse model, 50 µg of plasmids expressing YAP5SA and 10 µg plasmids expressing PB transposase were diluted in sterile Ringer's solution with a volume equal to 10% of body weight. The mixture was injected into HKDC1 KO and WT C57BL/6 mice via the tail vein within 5–7 s. For hepatic portal mouse model, Hepa1-6 cells ($3 \times 10^5$) with indicated phenotypes were injected through the portal vein into 6-week-old C57BL/6 mice. For hepatic portal mouse model, Hep55.1c cells ($4 \times 10^5$) with indicated phenotypes were injected through portal vein into 6 weeks old C57BL/6 mice. For Hepa1-6 xenograft mouse model, Hepa1-6 cells ($1 \times 10^6$) with indicated phenotypes were injected subcutaneously into 6 weeks old C57BL/6 mice and tumors were measured every three days with vernier caliper one week later, the tumor volumes were calculated using the following equation: length (mm) × width (mm) × depth (mm) × 0.52.

### Statistics and reproducibility
All experimental data are presented as mean ± SD or mean ± SEM as stated from at least three independent experiments. Two-tailed unpaired Student's *t*-test, Pearson's correlation analysis, log-rank (Mantel-Cox) test, one-way ANOVA or two-way ANOVA were used to calculate *P*-values by GraphPad Prism (v 9.0). $P < 0.05$ indicates a significant difference.

### Reporting summary
Further information on research design is available in the Nature Portfolio Reporting Summary linked to this article.

## Data availability
The transcriptomic data used in this study are available in the HCC cases in The Cancer Genome Atlas (TCGA) database under accession code phs000178 [https://www.ncbi.nlm.nih.gov/projects/gap/cgi-bin/study.cgi?study_id=phs000178.v11.p8]. The GO30140 and IMbrave150 cohorts used in this study are available in the European Genome-

phenome Archive (EGA) under accession code EGAD00001008130 [https://web2.ega-archive.org/datasets/EGAD00001008130]. Mass spectrometry proteomics data generated in this study have been deposited to the ProteomeXchange Consortium database under accession code PXD047388 [https://www.iprox.cn/page/project.html?id=IPX0007661000]. The remaining data are available within the Article, Supplementary Information, or Source Datafile. Source data are provided in this paper.

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

## Acknowledgements
This work was supported in part by grants from the National Natural Science Foundation of China (82341013, 92357301, 82322050, 82130087, 81930083, 82192893, 82072656, U22A20156), the National Key R&D Program of China (2018YFA0800303, 2022YFA1304504).

## Author contributions
P.G. and X.Z. conceived and supervised this study. X.Z., Y.Z., X.Y. H.Z., and P.G. designed the experiments. Y.Z., M.W., L.Y., S.S., Y.X.Z., X.Q., T.Z., M.Y., Z.Y., J.C., X.M., S.Q., S.L, and R.L., and W.J. performed and analyzed the experiments. X.Z., Y.Z., and P.G. wrote the paper. All the authors read and approved the manuscript.

## Competing interests
The authors declare no competing interests.
