## [Peer Review File · Nature Communications]

HKDC1 promotes tumor immune evasion by coupling cytoskeleton to STAT1 activation and PD-L1 expressionREVIEWER COMMENTS

Reviewer #1 (Remarks to the Author): with expertise in HKDC1, liver physiopathology

Overall, this is a very good paper. Only one major concern noted.

One major concern is regarding the studies looking at how HKDC1 interacts with other proteins. Where was the flag tag placed on HKDC1? If on the N-terminus, it is concerning because the mitochondria targeting domain is on the N-terminus. This will impact the IF studies as well. I could not find these details in the paper

Fig 1b-needs better labeling, hard to understand what being shown

Fig 1b-c, is this the Yap model or HKDC1 KO, hard to follow in the fig legend

Fig 1 d, same issue, needs more precision in the description

For the authors to note in the paper if indicated, that HKDC1 hexokinase activity is known to be relatively low, less than GCK.

Where all the IF images scored as blinded?

I struggled to understand Fig 5 overall.

Reviewer #2 (Remarks to the Author): with expertise in cancer immunology, PD-L1 regulation

This is a very interesting and comprehensive study. The authors identified a new role of HKDC1 in regulating immune evasion via STAT1 activation and suggested a new combination strategy to enhance anti-tumor immune response. It could lead to the development of therapeutic strategies for HCC. There are a few things that can be done to enhance the feasibility and reliability:

1. In Figure 1, was there any significant change in the Treg population? The authors need to

show and discuss additional data regarding Treg's role in addition to CD8 T cells.

2. In Figures 3 and 4, all of the IP/WB data were obtained from the overexpressed proteins exogenously. Thus, it remains the question of whether these proteins' interactions exist in a physiological condition without overexpression of proteins.

3. In Figure 5, how does the VNPSiHKDC1 precisely deliver to the tumor cell? Is there any unwanted effect of the VNPSiHKDC1 on other cell types, in particular immune cells? In Fig 2i and 2j, in addition to PD-L1, many other immune receptors and ligands' expression were modulated by HKDC1 KD or KO.

4. Because both siHKDC1 and anti-PD-L1 target the same molecule, PD-L1, to improve a combination treatment synergistically, a combination of siHKDC1 and other ICB antibodies (i.e., anti-PD-1) would be more reasonable. Is there a rationale for combining siHKDC1 and anti-PD-L1 instead of anti-PD-1?

5. The CD8+ T cell killing assay needs to be performed in additional HCC cells with no HBV protein expression. Hep3B may have higher immunogenicity than other HCC cell lines because Hep3B cells express HBV proteins.

6. No data shows the expression of mHKDC1 protein in the Hepa1-6 mHKDC1 KD cells.

Reviewer #3 (Remarks to the Author): with expertise in HCC, cancer immunology

There is limited number of reports suggest that HKDC1 regulates tumor development. In this manuscript, the authors found that in HCC patients HKDC1 expression is increased in tumors compared to surrounding tissues. HKDC1 deficiency suppressed HCC development in mice. Interestingly, knocking down HKDC1 in tumor line Hep1-6 reduced in vivo tumor growth which is CD8+T cell dependent. Consistently, tumor infiltrating CD8+T cells showed less exhaustion in tumors with HKDC1 knockdown. Then, the authors found the tumor regulation function of HKDC1 is independent of its hexokinase activity. Interestingly, in vitro studies showed that HKDC1 induces PD-L1 expression via ACTA2/STAT1 signaling. Based on

these results the authors claim that HKDC1 promotes tumor immune evasion via upregulation of PD-L1. The observation that tumoral HKDC1 suppresses the anti-tumor function of CD8+T cells is interesting. However, the authors did not provide convincing in vivo data to support that the effect is indeed mediated by PD-L1. In fact, the claim of PD-L1 dependent mechanism causes many confusions/troubles. It is difficult to explain why anti-HKDC1 can help anti-PD-L1 therapy if both act to block PD-L1 signal, especially the authors claimed a “synergistic effect”.

Major concerns:

- (1) If the authors want to claim the PD-L1 dependent immune evasion by HKDC1 as in the title, then clear in vivo evidence must be provided. Such as using a PD-L1 KO tumor to test if modulating HKDC1 can still influence in vivo tumor growth.
- (2) If HKDC1 acts only through PD-L1, the claim of “synergistic effect” with anti-PD-L1 is not valid and should be removed. I don't see the potential clinical benefit to further block PD-L1 by HKDC1 inhibitor on top of anti-PD-L1 therapy, which is hard to argue by the potential insufficient dose of anti-PD-L1.
- (3) PD-L1 is not only expressed by tumor but also by tumor stromal cells especially macrophages. And the importance of macrophage PD-L1 expression in tumor progression including HCC has been reported by many groups include a recent Gut paper (PMID 35173040). The authors should test the effect of targeting non-tumor HKDC1 on tumor progression (such as using tumor expressing knockdown resistant HKDC1)
- (4) Hepa1-6 tumor line is derived from a C57L mouse, which is not a true syngeneic tumor model and has minor MHC mismatch when used in C57BL/6 mice. A true syngeneic HCC model should be used to repeat the key findings.
- (5) The authors never studied PD-L1 activation, and the title is misleading.
- (6) The in vitro CD8+T cell cytotoxicity results are hard to believe. The anti-tumor CD8+ T cytotoxicity is well known to be antigen specific. How many naive mouse CD8+ T cells can recognize Hep1-6 tumor cells? It is much more problematic for assay using human Hep3B, which proper donor T cells are needed to ensure the TCR/MHC-I recognition in the first place, not to mention the presence of Hep3B recognizing CD8+ T cells.
- (7) The shmHKDC1-1 targets coding region of HKDC1, why it did not knockdown the HKDC1 expression from vectors such as in Fig.2?

Point-by-point response to the comments

We thank the reviewers for the insightful comments and constructive suggestions that have helped us with the preparation of a revised manuscript. Over the past three months, we have performed additional experiments to address all the concerns and comments raised by our reviewers. Here we submitted a substantially improved manuscript along with our point-by-point response. For the reviewers' convenience, we have appended all the related figures in this file, which we labeled as **Fig. R1** to **Fig. R8**.

Reviewer #1 (Remarks to the Author): with expertise in HKDC1, liver physiopathology.

Comments 1-1:

Overall, this is a very good paper. Only one major concern noted.

Response: We are grateful for the positive comments provided by our reviewer.

Comments 1-2:

One major concern is regarding the studies looking at how HKDC1 interacts with other proteins. Where was the flag tag placed on HKDC1? If on the N-terminus, it is concerning because the mitochondria targeting domain is on the N-terminus. This will impact the IF studies as well. I could not find these details in the paper.

Response: We appreciate the reviewer's concern. This is an important point. We had the same consideration and placed the Flag-tag on the C-terminus of HKDC1. Following the reviewer's suggestion, we have supplied this information in the figure legend of **Fig. 2m in the revised manuscript**.

Comments 1-3:

Fig 1b-needs better labeling, hard to understand what being shown.

Response: Thank you for pointing this out. Following the reviewer's suggestion, we have improved the labeling in **Fig. 1b in the revised manuscript**. To investigate whether HKDC1 could promote tumor progression by regulating antitumor immunity, we employed a spontaneous mouse model of HCC by injecting YAP5SA plasmid into mouse^{1,2} and examined the composition of tumor-infiltrating immune cell populations in YAP5SA-induced HCC mouse model of wild-type (WT) and HKDC1 knockout (KO) mice by flow cytometry. T-distributed stochastic neighbor embedding (t-SNE) plot and the bar graph showed the percentage of tumor-infiltrating PD1⁺CD8⁺ T cells was significantly lower in HKDC1 KO mice than that in the WT group (**Fig. 1b in the original manuscript**), suggesting that loss of HKDC1 in cancer cells might enhance the antitumor activity of tumor-infiltrating CD8⁺ T cells.

Comments 1-4:

Fig 1b-c, is this the Yap model or HKDC1 KO, hard to follow in the fig legend.

Response: We appreciate the reviewer for the critical comments. By employing a spontaneous HCC mouse model^{1,2}, we injected YAP5SA plasmid into WT and HKDC1 KO mice and observed that HKDC1 knockout significantly suppressed HCC incidence and development (**Extended Data Fig. 1d, e in the original manuscript**, see also as **Extended Data Fig. 1d, e in the revised manuscript**). More importantly, the flow-cytometric results revealed that the percentage of tumor-infiltrating PD1⁺CD8⁺ T cells was significantly lower in HCC-bearing HKDC1 KO mice than that in the tumor-bearing WT group (**Fig. 1b in the original manuscript**, see also as **Fig. 1b in the revised manuscript**). Moreover, the proportion of exhausted CD8⁺ T cells (*i.e.*, PD-1⁺ or LAG-3⁺) was significantly decreased and CD8⁺ T cell activity (reflected by the proportion of IFN γ ⁺ or Granzyme B⁺ cells) was obviously increased in HCC-bearing HKDC1 KO mice compared to that in tumor-bearing WT mice (**Fig. 1c, d in the original manuscript**, see also as **Fig. 1c, d in the revised manuscript**). These data suggested that loss of HKDC1 in cancer cells enhances the antitumor activity of tumor-infiltrating CD8⁺ T cells. Following our reviewer's suggestion, we have improved the Results section and figure legends in the revised manuscript.

Comments 1-5:

Fig 1d, same issue, needs more precision in the description.

Response: We appreciate the reviewer's suggestion and have improved the figure legend in the revised manuscript.

Comments 1-6:

For the authors to note in the paper if indicated, that HKDC1 hexokinase activity is known to be relatively low, less than GCK.

Response: We appreciate the reviewer for the important information. Actually, we also noticed that according to the study of Pusec et al.³, HKDC1 has low glucose-phosphorylating ability, and discussed this important feature of HKDC1 in the Introduction section in the original manuscript.

Comments 1-7:

Where all the IF images scored as blinded?

Response: We appreciate the concerns raised by the reviewer. The investigators were not blinded to allocation during IF experiments and data analysis. However, all the experiments were reproduced independently by different individuals.

Comments 1-8:

I struggled to understand Fig 5 overall.

Response: We apologize for the confusion. Following the reviewers' suggestion (also suggested by other reviewers), we have performed additional *in vivo* experiments to validate the antitumor effect of HKDC1 inhibition incorporating with anti-PD-1. In an orthotopic tumor mouse model by inoculating Hepa1-6 cells via the hepatic portal vein in C57BL/6 immune-competent mice, we observed that HKDC1 inhibition by treating mice with VNP^{siHKDC1/MTO} (siHKDC1 hereafter) and PD-1 antibodies (α PD-1) each inhibited tumor growth, while siHKDC1 further enhanced the antitumor effects of α PD-1 (**Fig. R1a**, see also as **Fig. 5a in the revised manuscript**). qPCR analysis

confirmed that siHKDC1 significantly suppressed *Cd274* expression (**Fig. R1b**, see also as **Fig. 5b in the revised manuscript**). Furthermore, flow-cytometric analysis indicated that the combination of siHKDC1 and α PD-1^{low} resulted in significant suppression of exhaustion in tumor-infiltrating CD8⁺ T cells, which displayed strikingly increased antitumor activity (**Fig.R1c, d**, see also as **Fig. 5d, e in the revised manuscript**). Notably, the combination treatment of siHKDC1 and α PD-1^{low} (25 μ g/mouse) exhibited similar antitumor effects with that of α PD-1^{high} (250 μ g/mouse) (**Fig. R1a, b**, see also as **Fig. 5a, b in the revised manuscript**); however, median survival time was longer while the relative risk of animal immune-related adverse events (irAEs)⁴⁻⁶ was lower in the combination treatment group compared to that in α PD-L1^{high} group mice (**Fig. R1e-j**, see also as **Fig. 5c and Extended Data Fig. 6a-e in the revised manuscript**). Collectively, these results clearly demonstrated that HKDC1 inhibition enhances CD8⁺ T cell-mediated antitumor immunity by regulating PD-L1 expression and HKDC1 has the potential to become an important target for combinational immunotherapies. To avoid confusion, now we have replaced the original results of *in vivo* experiments in the revised manuscript with this new data (**Fig. 5 and Extended Data Fig. 6 in the revised manuscript**).

Fig. R1. HKDC1 inhibition incorporating with PD-1 blockade enhances T cell antitumor response in HCC model mice. **a.** The H.E. quantification of tumor area/liver area (mm^2/mm^2) in hepatic portal mouse model with indicated genotypes ($n = 6$ male mice per group) is displayed on the right. Hepa1-6 cells were injected through hepatic portal vein into C57BL/6 mice (6-8 weeks), and randomly divided into five groups with indicated treatment. VNP^{Ctrl/MTO} or VNP^{siHKDC1/MTO} (0.34 OD/per mice) was injected through tail vein, then anti-PD-1 antibodies (α PD-1) or IgG2b isotype control was injected intraperitoneally twice a week. Images of liver and representative H.E. staining are displayed on the left. Scale bars, 1 mm. **b.** qPCR analysis of mRNA levels of *Hkdc1* and *Cd274* in the hepatic portal mouse model in (a). **c.** The percentage of tumor-infiltrating PD-1⁺ or LAG-3⁺ CD8⁺ T cells in hepatic portal mouse model in (a) was analyzed by flow cytometry. **d.** The percentage of tumor-infiltrating IFN γ ⁺ or GzmB⁺ CD8⁺ T cells in hepatic portal mouse model in (a) was analyzed by flow cytometry. **e.** Overall survival of the hepatic portal mouse model in (a). **f.** Body weight of hepatic portal mouse model in (a). **g-i.** RBC, HCT and HGB in peripheral blood of the hepatic portal mouse model in (a) were measured before sacrifice. **j.** Levels of blood glucose of the hepatic portal mouse model in (a) were measured before sacrifice. Data are presented as mean \pm SD (a-j). *P*-values were calculated by one-way ANOVA (a-d, g-j), log-rank (Mantel-Cox) test (e), or two-way ANOVA (f).

Reviewer #2 (Remarks to the Author): with expertise in cancer immunology, PD-L1 regulation.

Comments 2-1:

This is a very interesting and comprehensive study. The authors identified a new role of HKDC1 in regulating immune evasion via STAT1 activation and suggested a new combination strategy to enhance anti-tumor immune response. It could lead to the development of therapeutic strategies for HCC. There are a few things that can be done to enhance the feasibility and reliability:

Response: We appreciate our reviewer for positive opinion of our work and the insightful suggestions that help us enhance the quality of this study.

Comments 2-2:

In Figure 1, was there any significant change in the Treg population? The authors need to show and discuss additional data regarding Treg's role in addition to CD8 T cells.

Response: We thank the reviewer for the insightful comments. To address this point, we further examined the population of Treg cells from YAP5SA-induced HCC mouse model of wild-type (WT) and HKDC1 knockout (KO) mice, and our results showed no significant change in the tumor-infiltrating Treg population in HKDC1 KO mice than that in the WT group (**Fig. R2a**). Then, we employed another orthotopic HCC mouse model established by hepatic portal vein injection of Hepa1-6 murine liver cancer cells expressing short hairpin RNA targeting HKDC1 (shmHKDC1s) or Non-targeted control (NTC) and examined the percentage of tumor-infiltrating Treg cells. Our results showed that HKDC1 knockdown in cancer cells had no effect on the percentage of Treg cells in Hepa1-6 orthotopic tumor (**Fig. R2b**). Moreover, similar results were observed in Hep55.1c orthotopic tumor-bearing mouse models (**Fig. R2c**). Liver Treg cells play a crucial role in maintaining the peripheral tolerance state of liver. In addition, it has been reported that increased Treg cells correlated with impaired CD8⁺ T cell activity in HCC patients and HCC immunosurveillance⁷. Our

results indicated that aberrant HKDC1 expression in cancer cells increased exhaustion of tumor-infiltrating CD8⁺ T cells while having no obvious effect on the population of Treg cells in HCC microenvironment. Thus, in current study, we did not further investigate whether HKDC1 depletion in cancer cells had any effect on the percentage of some Treg cell subpopulations. Considering the crucial role of Treg cells and the novel function of HKDC1 in regulating tumor immunity, the potential correlation between tumor intrinsic HKDC1 expression and Treg cell function warrants further independent study.

Fig. R2. HKDC1 depletion in cancer cells has no effect on the percentage of tumor-infiltrating Treg cells. **a.** The percentage of Treg cells in YAP5SA-induced HCC mouse model of wild-type (WT) and HKDC1 KO (HKDC1^{-/-}) mice was analyzed by flow cytometry (n = 4 male mice per group). **b.** The percentage of Treg cells in hepatic portal mouse model was analyzed by flow cytometry. Hepa1-6 cells with indicated genotypes were injected through hepatic portal vein into 6-8 weeks old C57BL/6 mice (n = 3 male mice per group). **c.** The percentage of tumor-infiltrating immune cell populations in hepatic portal mouse model was analyzed by flow cytometry. Hep55.1c cells with indicated genotypes were injected through hepatic portal vein into 6-8 weeks old C57BL/6 mice (n = 8 male mice per group). Data are presented as mean ± SD (a-c). P-values were calculated by two-tailed unpaired Student's *t*-test (a) or one-way ANOVA (b, c).

Comments 2-3:

In Figures 3 and 4, all of the IP/WB data were obtained from the overexpressed proteins exogenously. Thus, it remains the question of whether these proteins' interactions exist in a physiological condition without overexpression of proteins.

Response: We appreciate the reviewer for the critical comments. Following our reviewer's suggestion, we performed additional IP assays using antibodies targeting HKDC1 or STAT1 in HCC cells to confirm these proteins' interactions in a

physiological condition. The results showed that HKDC1 could associate with STAT1 (**Fig. R3a**, see also as **Fig. 3b** and **Extended Data Fig. 4a** in the revised manuscript) and ACTA2 (**Fig. R3b**, see also as **Fig. 4c** in the revised manuscript). More importantly, the interaction between STAT1 and ACTA2 was greatly attenuated under HKDC1 knockdown, while ACTA2 knockdown had no apparent effect on interaction between HKDC1 and STAT1 (**Fig. R3c, d**, see also as **Extended Data Fig. 5g, h** in the revised manuscript). These data demonstrated that HKDC1 is required for the interaction between STAT1 and ACTA2.

Fig. R3. HKDC1 is required for the interaction between STAT1 and ACTA2. **a.** Immunoprecipitation assay of interaction between endogenous HKDC1 and STAT1 in Hep3B (left) and HepG2 (right) cells. **b.** Immunoprecipitation assay of interaction between endogenous HKDC1 and ACTA2 in Hep3B cells. **c.** Immunoprecipitation assay of interaction between endogenous ACTA2 and STAT1 in Hep3B cells expressing NTC or shHKDC1. **d.** Immunoprecipitation assay of interaction between endogenous HKDC1 and STAT1 in Hep3B cells expressing NTC or shACTA2.

Comments 2-4:

In Figure 5, how does the VNPsiHKDC1 precisely deliver to the tumor cell? Is there

any unwanted effect of the VNPsiHKDC1 on other cell types, in particular immune cells? In Fig 2i and 2j, in addition to PD-L1, many other immune receptors and ligands' expression were modulated by HKDC1 KD or KO.

Response: We appreciate the reviewer for the concern regarding the unwanted effects of VNP^{siHKDC1/MTO}. VNP^{siHKDC1/MTO} is a poly lactic-co-glycolic acid (PLGA)-based vesicle-like nanoparticle that simultaneously delivers siRNA targeting HKDC1 and mitoxantrone hydrochloride (MTO). PLGA is one of the most promising candidates in biomedical-related field because of its excellent performance in biocompatibility, reduction of side effects and so on. Actually, in our previous study⁸, we found that the nanoparticles were also enriched in the liver and kidney, apart from the tumor tissues in the tumor-bearing mice injecting VNP^{siRNA/MTO}. To address the reviewer's concern, we examined the expression levels of HKDC1 in tumor cells and other cells in the liver from YAP5SA-induced HCC mouse model of wild-type (WT) mice. Relative expression analysis by qPCR indicated that HKDC1 was expressed at substantially higher levels in tumor cells compared to that in other CD45⁻ cells and immune cells including B cells, macrophages and so on (**Fig. R4**, see also as **Extended Data Fig. 2a in the revised manuscript**).

Fig. R4. The expression of HKDC1 is substantially higher in tumor cells compared to that in other CD45⁻ cells and immune cells. qPCR analysis of *Hkdc1* mRNA levels in indicated cells in YAP5SA-induced HCC mouse model of WT mice (n = 3 male mice). Data are presented as mean ± SD. P-values were calculated by one-way ANOVA.

It was worth noting that the percentage of tumor-infiltrating PD1⁺CD8⁺ T cells was significantly lower in YAP5SA-induced HCC-bearing HKDC1 KO mice than that in the WT group, while HKDC1 depletion had no effect on the percentages of other

immune cells (**Fig. 1b in the original manuscript**, see also as **Fig. 1b in the revised manuscript**). In order to investigate whether the decreased exhaustion and increased activity of tumor-infiltrating T cells in HCC-bearing HKDC1 KO mice are caused by the effects of HKDC1 depletion specifically in tumor cells or immune cells, we used an HCC mouse model established by hepatic portal vein injection of murine liver cancer cells expressing shmHKDC1s or NTC and quantified exhaustion and activity of tumor-infiltrating CD8⁺ T cells. Flow-cytometric results revealed that the infiltrated CD8⁺ T cells exhibited lower PD-1 and LAG-3 expression and higher IFN γ and GzmB expression in mice inoculated with mHKDC1 KD Hepa1-6 cells compared with those in NTC-inoculated mice (**Extended Data Fig. 1g, h in the original manuscript**, also see as **Extended Data Fig. 2c, d in the revised manuscript**), suggesting that loss of HKDC1 in cancer cells enhanced the antitumor activity of tumor-infiltrating CD8⁺ T cells. Collectively, although VNP^{siHKDC1/MTO} might be enriched in the liver other than tumor cells, HKDC1 depletion had little effect on other cells, probably because of the substantially lower expression levels of HKDC1 in these cells. These data further highlighted that HKDC1 is a potential target for combinational immunotherapies.

As the reviewer has pointed out, in addition to PD-L1, the expression levels of some other immune checkpoint markers were increased in the tumor tissues of YAP5SA-induced HCC mouse model compared to that in the noncancerous liver tissues, which was significantly suppressed in the tumor tissues of HKDC1 KO mice (**Fig. 2j in the original manuscript**, see also as **Fig. 2j in the revised manuscript**). This is probably caused by the comprehensive *in vivo* conditions, especially that HKDC1 KO significantly suppressed tumor incidence and growth (**Extended Data Fig. 1d, e in the original manuscript**, see also as **Extended Data Fig. 1d, e in the revised manuscript**). Therefore, we detected the mRNA levels of immune checkpoint markers in IFN γ -stimulated Hep3B cells expressing shHKDC1s or NTC. Relative expression analysis by qPCR indicated that HKDC1 knockdown significantly reduced the *CD274* (encoding PD-L1) expression elevated by IFN γ stimulation in Hep3B cells (**Fig. 2i in the original manuscript**, see also as **Fig. 2i in the revised manuscript**).

Notably, the transcription of other ICB markers had no significant change in HKDC1 knockdown cells compared to that in NTC cells (**Fig. 2i in the original manuscript**, see also as **Fig. 2i in the revised manuscript**). Furthermore, qPCR analysis revealed a strong correlation between *HKDC1* and *CD274* expression in clinical HCC tissues (**Fig. 2n in the original manuscript**, see also as **Fig. 2n in the revised manuscript**). These results indicated that HKDC1 enhanced PD-L1 transcription in HCC cells. Moreover, using HCC mouse models established by hepatic portal vein injection of murine liver cancer cells with indicated phenotypes, we found that tumor growth was promoted in mice inoculated murine liver cancer cells with mHKDC1 overexpressing (mHKDC1), but mPD-L1 knockout (sgmPD-L1) attenuated tumor growth promoted by HKDC1 overexpression (**Fig. R5a, b**, see also as **Extended Data Fig. 3k, l in the revised manuscript**). These results suggested that HKDC1 promotes tumor progression, and this promotion is PD-L1 dependent. Collectively, our data emphasized the pivotal role of PD-L1 in HKDC1-stimulated tumor development.

Fig. R5. HKDC1 promotes HCC progression in a PD-L1-dependent way. **a.** Image of livers in Hepa1-6 xenografts with indicated genotypes ($n = 5$ mice per group) is shown on the left. Hepa1-6 cells with indicated genotypes were injected through hepatic portal vein into C57BL/6 mice (6-8 weeks). Compared liver/body weight ratio is shown on the right. **b.** Image of livers in Hep55.1c xenografts with indicated genotypes ($n = 5$ mice per group) is shown on the left. Hep55.1c cells with indicated genotypes were injected through hepatic portal vein into C57BL/6 mice (6-8 weeks). Compared liver/body weight ratio is shown on the right. Data are presented as mean \pm SD (a, b). P -values were calculated by two-tailed unpaired Student's t -test (a, b).

Comments 2-5:

Because both siHKDC1 and anti-PD-L1 target the same molecule, PD-L1, to improve a combination treatment synergistically, a combination of siHKDC1 and other ICB antibodies (i.e., anti-PD-1) would be more reasonable. Is there a rationale for

combining siHKDC1 and anti-PD-L1 instead of anti-PD-1?

Response: We appreciate the reviewer for the insightful comments and constructive suggestions. Following the reviewers' suggestion, we have performed additional *in vivo* experiments to validate the antitumor effect of HKDC1 inhibition incorporating with anti-PD-1. Actually, the reviewer #1 also raised a similar question. For your convenience, we append the figures here once again (**Fig. R1**). In an orthotopic tumor mouse model by inoculating Hepa1-6 cells via the hepatic portal vein in C57BL/6 immune-competent mice, we observed that HKDC1 inhibition by treating mice with VNP^{siHKDC1/MTO} (siHKDC1 hereafter) and PD-1 antibodies (α PD-1) each inhibited tumor growth, while siHKDC1 further enhanced the antitumor effects of α PD-1 (**Fig. R1a**, see also as **Fig. 5a in the revised manuscript**). qPCR analysis confirmed that siHKDC1 significantly suppressed *Cd274* expression (**Fig. R1b**, see also as **Fig. 5b in the revised manuscript**). Furthermore, flow-cytometric analysis indicated that the combination of siHKDC1 and α PD-1^{low} resulted in significant suppression of exhaustion in tumor-infiltrating CD8⁺ T cells, which displayed strikingly increased antitumor activity (**Fig. R1c, d**, see also as **Fig. 5d, e in the revised manuscript**). Notably, the combination treatment of siHKDC1 and α PD-1^{low} (25 μ g/mouse) exhibited similar antitumor effects with that of α PD-1^{high} (250 μ g/mouse) (**Fig. R1a, b**, see also as **Fig. 5a, b in the revised manuscript**); however, median survival time was longer while the relative risk of animal immune-related adverse events (irAEs)⁴⁻⁶ was lower in the combination treatment group compared to that in α PD-1^{high} group mice (**Fig. R1e-j**, see also as **Fig. 5c and Extended Data Fig. 6a-e in the revised manuscript**). Collectively, these results clearly demonstrated that HKDC1 inhibition enhances CD8⁺ T cell-mediated antitumor immunity by regulating PD-L1 expression and HKDC1 has the potential to become an important target for combinational immunotherapies. To avoid confusion, now we have replaced the original results of *in vivo* experiments in the revised manuscript with this new data (**Fig. 5 and Extended Data Fig. 6 in the revised manuscript**).

Fig. R1. HKDC1 inhibition incorporating with PD-1 blockade enhances T cell antitumor response in HCC model mice. **a.** The H.E. quantification of tumor area/liver area (mm^2/mm^2) in hepatic portal mouse model with indicated genotypes ($n = 6$ male mice per group) is displayed on the right. Hepa1-6 cells were injected through hepatic portal vein into C57BL/6 mice (6-8 weeks), and randomly divided into five groups with indicated treatment. $\text{VNP}^{\text{Ctrl}/\text{MTO}}$ or $\text{VNP}^{\text{siHKDC1}/\text{MTO}}$ (0.34 OD/per mice) was injected through tail vein, then

anti-PD-1 antibodies (α PD-1) or IgG2b isotype control was injected intraperitoneally twice a week. Images of liver and representative H.E. staining are displayed on the left. Scale bars, 1 mm. **b.** qPCR analysis of mRNA levels of *Hkdc1* and *Cd274* in the hepatic portal mouse model in (a). **c.** The percentage of tumor-infiltrating PD-1⁺ or LAG-3⁺ CD8⁺ T cells in hepatic portal mouse model in (a) was analyzed by flow cytometry. **d.** The percentage of tumor-infiltrating IFN γ ⁺ or GzmB⁺ CD8⁺ T cells in hepatic portal mouse model in (a) was analyzed by flow cytometry. **e.** Overall survival of the hepatic portal mouse model in (a). **f.** Body weight of hepatic portal mouse model in (a). **g-i.** RBC, HCT and HGB in peripheral blood of the hepatic portal mouse model in (a) were measured before sacrifice. **j.** Levels of blood glucose of the hepatic portal mouse model in (a) were measured before sacrifice. Data are presented as mean \pm SD (a-j). *P*-values were calculated by one-way ANOVA (a-d, g-j), log-rank (Mantel-Cox) test (e), or two-way ANOVA (f).

Comments 2-6:

The CD8⁺ T cell killing assay needs to be performed in additional HCC cells with no HBV protein expression. Hep3B may have higher immunogenicity than other HCC cell lines because Hep3B cells express HBV proteins.

Response: Following the reviewer's suggestion, we performed additional CD8⁺ T cell killing assay in HepG2 cells. As a result, HKDC1 KD could sensitize HepG2 cells to cytolysis by CD8⁺ T cells (**Fig. R6a**), and notably, overexpression of either wild type HKDC1 (HKDC1^{WT}) or catalytic site mutant HKDC1^{S602A} in HKDC1 KD HepG2 cells could restore resistance to CD8⁺ T cell cytolytic activity (**Fig. R6b**), suggesting that HKDC1 expression in HCC cells could increase the resistance to CD8⁺ T cell-mediated cytolysis independently of its hexokinase function. To further investigate whether HKDC1 association with filament protein ACTA2 is necessary for its promotion of tumor immune evasion, we performed additional CD8⁺ T cell killing assay in HepG2 cells with HKDC1 overexpression and ACTA2 knockdown. The result showed that the increased resistance to CD8⁺ T cell-mediated cytolysis induced by HKDC1 overexpression could be reversed by ACTA2 KD in HepG2 cells (**Fig. R6c**). Collectively, these data suggested that HKDC1 expression could enhance tumor immune evasion dependent on its association with filament protein ACTA2, but uncoupling with its hexokinase function.

Fig. R6. HKDC1 enhances tumor immune evasion dependent on its association with filament protein ACTA2, but uncoupling with its hexokinase function. a-c. Cell death analysis of indicated HepG2 cells, co-cultured with pre-activated CD8⁺ T cells separated from human PBMCs, was analyzed by flow cytometry. The data are presented as mean \pm s.e.m of three independent experiments (a-c). *P*-values were calculated using one-way ANOVA (a-c).

Comments 2-7:

No data shows the expression of mHKDC1 protein in the Hepa1-6 mHKDC1 KD cells.

Response: We truly appreciate our reviewer for the critical comments. Actually, we did have tried several commercialized antibodies targeting HKDC1, but no commercialized antibody was available for detecting the protein level of mHKDC1. Therefore, we had to detect the expression levels of *Hkdcl* in murine cells by qRT-PCR.

Reviewer #3 (Remarks to the Author): with expertise in HCC, cancer immunology

Comments 3-1:

There is limited number of reports suggest that HKDC1 regulates tumor development. In this manuscript, the authors found that in HCC patients HKDC1 expression is increased in tumors compared to surrounding tissues. HKDC1 deficiency suppressed HCC development in mice. Interestingly, knocking down HKDC1 in tumor line Hepa1-6 reduced in vivo tumor growth which is CD8⁺T cell dependent. Consistently, tumor infiltrating CD8⁺T cells showed less exhaustion in tumors with HKDC1 knockdown. Then, the authors found the tumor regulation function of HKDC1 is independent of its hexokinase activity. Interestingly, *in vitro* studies showed that HKDC1 induces PD-L1 expression via ACTA2/STAT1 signaling. Based on these results the authors claim that HKDC1 promotes tumor immune evasion via upregulation of PD-L1. The observation that tumoral HKDC1 suppresses the anti-tumor function of CD8⁺T cells is interesting. However, the authors did not provide convincing in vivo data to support that the effect is indeed mediated by PD-L1. In fact, the claim of PD-L1 dependent mechanism causes many confusions/troubles. It is difficult to explain why anti-HKDC1 can help anti-PD-L1 therapy if both act to block PD-L1 signal, especially the authors claimed a “synergistic effect”.

Response: We thank the reviewer for the positive opinion and encouraging comments that have well summarized this study. Following the reviewer’s suggestion, we performed additional *in vivo* and *in vitro* experiments to support the conclusion of this study. For your convenience, we append the figures below and address your concern in detail.

Comments 3-2:

If the authors want to claim the PD-L1 dependent immune evasion by HKDC1 as in the title, then clear in vivo evidence must be provided. Such as using a PD-L1 KO

tumor to test if modulating HKDC1 can still influence *in vivo* tumor growth.

Response: We appreciate the reviewer for the comments and suggestions. Following your suggestion, we have performed additional experiments using two HCC mouse models established by hepatic portal vein injection of murine liver cancer cells with indicated phenotypes. And our results showed that tumor growth was significantly promoted under mHKDC1 overexpressing (mHKDC1), but mPD-L1 knockout (sgmPD-L1) attenuated tumor growth promoted by HKDC1 overexpression (**Fig. R5**, see also as **Extended Data Fig. 3k, l in the revised manuscript**). These results suggested that HKDC1 promotes tumor progression, and this promotion is PD-L1 dependent.

In addition, our results revealed that the expression of PD-L1 was significantly reduced in tumors from YAP5SA-induced HCC-bearing HKDC1 knockout (KO) mice than that in the wild-type (WT) group (**Fig. 2l in the original manuscript**, see also as **Fig. 2l in the revised manuscript**). Indeed, qPCR analysis of *Cd274* (encoding PD-L1) expression in tumor tissues from xenograft model mice generated by subcutaneous inoculation with Non-targeted control (NTC) or mHKDC1 knockdown (KD) Hepa1-6 cells overexpressing mHKDC1^{WT} or mHKDC1^{S600A} showed that *Cd274* mRNA levels were significantly reduced in mHKDC1 KD tumors compared to that in NTC tumors, and the decline could be rescued by mHKDC1^{WT} or mHKDC1^{S600A} overexpression (**Extended Data Fig. 2j in the original manuscript**, see also as **Extended Data Fig. 3j in the revised manuscript**). Furthermore, qPCR analysis revealed a strong correlation between *HKDC1* and *CD274* expression in clinical HCC tissues (**Fig. 2n in the original manuscript**, see also as **Fig. 2n in the revised manuscript**). These results indicated that HKDC1 enhanced PD-L1 transcription level. Taken together, our results showed that HKDC1 promotes *in vivo* tumor progression by up-regulating PD-L1.

Fig. R5. HKDC1 promotes HCC progression in a PD-L1-dependent way. **a.** Image of livers in Hep55.1c xenografts with indicated genotypes ($n = 5$ mice per group) is shown on the left. Hep55.1c cells with indicated genotypes were injected through hepatic portal vein into C57BL/6 mice (6-8 weeks). Compared liver/body weight ratio is shown on the right. **b.** Image of livers in Hepa1-6 xenografts with indicated genotypes ($n = 5$ mice per group) is shown on the left. Hepa1-6 cells with indicated genotypes were injected through hepatic portal vein into C57BL/6 mice (6-8 weeks). Compared liver/body weight ratio is shown on the right. Data are presented as mean \pm SD (a, b). P -values were calculated by two-tailed unpaired Student's t -test (a, b).

Comments 3-3:

If HKDC1 acts only through PD-L1, the claim of “synergistic effect” with anti-PD-L1 is not valid and should be removed. I don't see the potential clinical benefit to further block PD-L1 by HKDC1 inhibitor on top of anti-PD-L1 therapy, which is hard to argue by the potential insufficient dose of anti-PD-L1.

Response: We apologize for not making the statement clear and have now improved the writing. Following your suggestion (also suggested by other reviewers), we have performed additional *in vivo* experiments to validate the antitumor effect of HKDC1 inhibition incorporating with anti-PD-1. In an orthotopic tumor mouse model by inoculating Hepa1-6 cells via the hepatic portal vein in C57BL/6 immune-competent mice, we observed that HKDC1 inhibition by treating mice with VNP^{siHKDC1/MTO} (siHKDC1 hereafter) and PD-1 antibodies (α PD-1) each inhibited tumor growth, while siHKDC1 further enhanced the antitumor effects of α PD-1 (**Fig. R1a**, see also as **Fig. 5a in the revised manuscript**). qPCR analysis confirmed that siHKDC1 significantly suppressed *Cd274* expression (**Fig. R1b**, see also as **Fig. 5b in the revised manuscript**). Furthermore, flow-cytometric analysis indicated that the combination of siHKDC1 and α PD-1^{low} resulted in significant suppression of exhaustion in tumor-infiltrating CD8⁺ T cells, which displayed strikingly increased

antitumor activity (**Fig.R1c, d**, see also as **Fig. 5d, e in the revised manuscript**). Notably, the combination treatment of siHKDC1 and α PD-1^{low} (25 μ g/mouse) exhibited similar antitumor effects with that of α PD-1^{high} (250 μ g/mouse) (**Fig. R1a, b**, see also as **Fig. 5a, b in the revised manuscript**); however, median survival time was longer while the relative risk of immune-related adverse events (irAEs)⁴⁻⁶ was lower in the combination treatment group compared to that in α PD-1^{high} group mice (**Fig. R1e-j**, see also as **Fig. 5c and Extended Data Fig. 6a-e in the revised manuscript**). Collectively, these results clearly demonstrated that HKDC1 inhibition enhances CD8⁺ T cell-mediated antitumor immunity by regulating PD-L1 expression and HKDC1 has the potential to become an important target for combinational immunotherapies. To avoid confusion, now we have replaced the original results of *in vivo* experiments in the revised manuscript with this new data (**Fig. 5 and Extended Data Fig. 6 in the revised manuscript**).

Fig. R1. HKDC1 inhibition incorporating with PD-1 blockade enhances T cell antitumor response in HCC model mice. **a.** The H.E. quantification of tumor area/liver area (mm^2/mm^2) in hepatic portal mouse model with indicated genotypes ($n = 6$ male mice per group) is displayed on the right. Hepa1-6 cells were injected through hepatic portal vein into C57BL/6 mice (6-8 weeks), and randomly divided into five groups with indicated treatment. $\text{VNP}^{\text{Ctrl}/\text{MTO}}$ or $\text{VNP}^{\text{siHKDC1}/\text{MTO}}$ (0.34 OD/per mice) was injected through tail vein, then anti-PD-1 antibodies (α PD-1) or IgG2b isotype control was injected intraperitoneally twice a week. Images of liver and representative H.E. staining are displayed on the left. Scale bars, 1 mm. **b.** qPCR analysis of mRNA levels of *Hkdc1* and *Cd274* in the hepatic portal mouse model in (a). **c.** The percentage of tumor-infiltrating PD-1⁺ or LAG-3⁺ CD8⁺ T cells in hepatic portal mouse model in (a) was analyzed by flow cytometry. **d.** The percentage of tumor-infiltrating IFN γ ⁺ or GzmB⁺ CD8⁺ T cells in hepatic portal mouse model in (a) was analyzed by flow cytometry. **e.** Overall survival of the hepatic portal mouse model in (a). **f.** Body weight of hepatic portal mouse model in (a). **g-i.** RBC, HCT and HGB in peripheral blood of the hepatic portal mouse model in (a) were measured before sacrifice. **j.** Levels of blood glucose of the hepatic portal mouse model in (a) were measured before sacrifice. Data are presented as mean \pm SD (a-j). *P*-values were calculated by one-way ANOVA (a-d, g-j), log-rank (Mantel-Cox) test (e), or two-way ANOVA (f).

Comments 3-4:

PD-L1 is not only expressed by tumor but also by tumor stromal cells especially macrophages. And the importance of macrophage PD-L1 expression in tumor progression including HCC has been reported by many groups include a recent Gut paper (PMID 35173040). The authors should test the effect of targeting non-tumor HKDC1 on tumor progression (such as using tumor expressing knockdown resistant

HKDC1).

Response: We appreciate the reviewer for the insightful comments. To address the reviewer's concern, we examined the expression levels of HKDC1 in tumor cells and other cells in the liver from YAP5SA-induced WT mice. Relative expression analysis by qPCR indicated that HKDC1 was expressed at substantially higher levels in tumor cells compared to that in other CD45⁻ cells and immune cells including B cells, macrophages and so on (**Fig. R4**, see also as **Extended Data Fig. 2a in the revised manuscript**, the reviewer #2 also raised a similar question. For your convenience, we append the figures here once again). Actually, using an HCC mouse model established by injecting YAP5SA plasmid into WT and HKDC1 knockout mice, we examined the composition of tumor-infiltrating immune cell populations in YAP5SA-induced HCC-bearing mice and found that HKDC1 depletion had no effect on the percentages of other immune cell populations, but only significantly reduced the percentage of tumor-infiltrating PD1⁺CD8⁺ T cells (**Fig. 1b in the original manuscript**, see also as **Fig. 1b in the revised manuscript**). Collectively, our data indicated that HKDC1 depletion had little effect on other cells including the immune cells, probably because of the substantially lower expression levels of HKDC1 in these cells. These data further highlight that HKDC1 is a potential target for combinational immunotherapies.

Fig. R4. The expression of HKDC1 is substantially higher in tumor cells compared to that in other CD45⁻ cells and immune cells. qPCR analysis of *Hkdc1* mRNA levels in indicated cells in YAP5SA-induced HCC mouse model of WT mice (n = 3 male mice). Data are presented as mean ± SD. P-values were calculated by one-way ANOVA.

Comments 3-5:

Hepal-6 tumor line is derived from a C57L mouse, which is not a true syngeneic tumor model and has minor MHC mismatch when used in C57BL/6 mice. A true syngeneic HCC model should be used to repeat the key findings.

Response: We thank the reviewer for the constructive comments. Actually, the *in vivo* tumor models we employed in this study have been used previously in the studies involved in HCC and tumor immunology⁹⁻¹⁴. Following our reviewer's suggestion, we performed additional *in vivo* experiments using murine liver cancer cell line Hep55.1c that is derived from C57BL/6 mice. We inoculated Hep55.1c cells expressing shmHKDC1s or NTC by hepatic portal vein injection and quantified exhaustion and activity of tumor-infiltrating CD8⁺ T cells. As a result, tumor growth was suppressed in mice inoculated with mHKDC1 knockdown Hep55.1c cells (**Fig. R7a**, also see as **Extended Data Fig. 2e in the revised manuscript**). Furthermore, the infiltrated CD8⁺ T cells exhibited lower PD-1 and LAG-3 expression and higher IFN γ and GzmB expression in mice inoculated with mHKDC1 knockdown Hep55.1c cells compared with those in NTC-inoculated mice (**Fig. R7b, c**, also see as **Extended Data Fig. 2f, g in the revised manuscript**). Collectively with the results using YAP5SA-induced HCC mouse model of WT and HKDC1 KO mice (**Fig. 1c, d in the original manuscript**, see also as **Fig. 1c, d in the revised manuscript**), our data indicated that aberrant HKDC1 expression in cancer cells promotes tumor immune evasion by increasing exhaustion of tumor-infiltrating CD8⁺ T cells.

Fig. R7. Aberrant HKDC1 expression in cancer cells promotes tumor immune evasion by increasing exhaustion of tumor-infiltrating CD8⁺ T cells. **a.** The H.E. quantification of tumor area/liver area (mm²/mm²) in Hep55.1c xenografts with indicated genotypes (n = 8 male mice per group) is shown on the right. Hep55.1c with indicated genotypes were injected through hepatic portal vein into C57BL/6 mice (6-8 weeks). Image of livers is shown on the left. Scale bars, 1mm. **b.** The percentage of tumor-infiltrating PD-1⁺ or LAG-3⁺ CD8⁺ T cells in the hepatic portal mouse model in (a) was measured by flow cytometry. **c.** The percentage of tumor-infiltrating IFN γ ⁺ or GzmB⁺ CD8⁺ T cells in the hepatic portal mouse model in (a) was measured by flow cytometry. Data are presented as mean \pm SD (a-c). *P*-values were calculated by one-way ANOVA (a) or two-tailed unpaired Student's *t*-test (b, c).

Comments 3-6:

The authors never studied PD-L1 activation, and the title is misleading.

Response: We appreciate the reviewer for the critical comments. To avoid confusion, now we have replaced the title in the revised manuscript with “HKDC1 promotes tumor immune evasion by coupling cytoskeleton to STAT1 activation and PD-L1 expression”.

Comments 3-7:

The *in vitro* CD8⁺T cell cytotoxicity results are hard to believe. The anti-tumor CD8⁺ T cytotoxicity is well known to be antigen specific. How many naive mouse CD8⁺ T cells can recognize Hep1-6 tumor cells? It is much more problematic for assay using human Hep3B, which proper donor T cells are needed to ensure the TCR/MHC-I recognition in the first place, not to mention the presence of Hep3B recognizing CD8⁺ T cells.

Response: We appreciate the reviewer for the concern regarding antigen specific. Actually, for the CD8⁺ T cell-mediated tumor cell-killing assays, CD8⁺ T cells used to co-culture with tumor cells were pre-activated with anti-CD3 and anti-CD28 antibodies for indicated days, which stimulated the killing activity of CD8⁺ T cells¹⁴⁻¹⁷. To further address this point, we performed additional experiments using Hepa1-6-OVA cells (Hepa1-6 cells stably overexpressing Ovalbumin) and OT-1 CD8⁺ T cells (CD8⁺ T cells were separated from OT1-C57BL/6 mice spleen and pre-activated by anti-CD3 and anti-CD28 antibodies). The results showed that mHKDC1 knockdown in Hepa1-6-OVA cells enhanced CD8⁺ T cell-mediated tumor cell killing activity *in vitro* (**Fig. R8a**). More importantly, overexpression of either mHKDC1^{WT} or mHKDC1^{S600A} in mHKDC1 knockdown Hepa1-6-OVA cells could restore resistance to CD8⁺ T cell cytolytic activity (**Fig. R8b**). These data are consistent with our previous results suggesting that HKDC1 expression in HCC cells could attenuate CD8⁺ T cell-mediated tumor cell killing activity *in vitro* independent of its hexokinase function. However, due to the limited conditions to establish human-specific *in vitro* co-culture system in our laboratory, we cannot complete the specific cytotoxicity experiment with human CD8⁺ T cells. We hope the clarification and additional data could address the concerns raised by the reviewer.

Fig. R8. HKDC1 expression in HCC cells could attenuate CD8⁺ T cell-mediated tumor cell killing *in vitro* independent of its hexokinase function. a, b. Cell death analysis of indicated Hepa1-6-OVA cells, co-cultured with pre-activated OT-1 CD8⁺ T cells, was analyzed by flow cytometry. Data are presented as mean \pm s.e.m of three independent experiments (a, b). *P*-values were calculated by one-way ANOVA (a, b).

Comments 3-8:

The shmHKDC1-1 targets coding region of HKDC1, why it did not knockdown the HKDC1 expression from vectors such as in Fig.2?

Response: We appreciate the reviewer for pointing this out. Actually, two short hairpin RNAs targeting the coding region and 3'-UTR of mHKDC1 transcripts, respectively, were used in our study. We knocked down HKDC1 with shRNA targeting 3'-UTR of HKDC1 transcripts, which silenced endogenous HKDC1 but not the exogenously expressing HKDC1 only containing coding sequence. We have described this point in detail in the Results section in the revised manuscript.

References

1. Shen, S.Y. *et al.* A miR-130a-YAP positive feedback loop promotes organ size and tumorigenesis. *Cell Res* **25**, 997-1012 (2015).
2. Zhang, T. *et al.* ENO1 suppresses cancer cell ferroptosis by degrading the mRNA of iron regulatory protein 1. *Nat Cancer* **3**,75-89 (2022).
3. Pusec, C.M. *et al.* Hepatic HKDC1 Expression Contributes to Liver Metabolism. *Endocrinology* **160**, 313-330 (2019).
4. Chuah, S. *et al.* Uncoupling immune trajectories of response and adverse events from anti-PD-1 immunotherapy in hepatocellular carcinoma. *J Hepatol* **77**, 683-694 (2022).
5. Zhu, A.X. *et al.* Molecular correlates of clinical response and resistance to atezolizumab in combination with bevacizumab in advanced hepatocellular carcinoma. *Nat Med* **28**, 1599-1611 (2022).
6. Morad, G., Helmink, B.A., Sharma, P. & Wargo, J.A. Hallmarks of response, resistance, and toxicity to immune checkpoint blockade. *Cell* **185**, 576 (2022).

7. Zheng, C.H. *et al.* Landscape of Infiltrating T Cells in Liver Cancer Revealed by Single-Cell Sequencing. *Cell* **169**, 1342-1356 (2017).
8. Zhang, Y.X. *et al.* A siRNA-Assisted Assembly Strategy to Simultaneously Suppress "Self" and Upregulate "Eat-Me" Signals for Nanoenabled Chemo-Immunotherapy. *ACS Nano* **15**, 16030-16042 (2021).
9. Xie, M. *et al.* FGF19/FGFR4-mediated elevation of ETV4 facilitates hepatocellular carcinoma metastasis by upregulating PD-L1 and CCL2. *J Hepatol* **79**, 109-125 (2023).
10. Wei, Y. *et al.* Plasma Cell Polarization to the Immunoglobulin G Phenotype in Hepatocellular Carcinomas Involves Epigenetic Alterations and Promotes Hepatoma Progression in Mice. *Gastroenterology* **156**, 1890-1904 (2019).
11. Li, J. *et al.* Co-inhibitory Molecule B7 Superfamily Member 1 Expressed by Tumor-Infiltrating Myeloid Cells Induces Dysfunction of Anti-tumor CD8⁺ T Cells. *Immunity* **48**, 773-786 (2018).
12. Wu, R.Q. *et al.* Immune checkpoint therapy-elicited sialylation of IgG antibodies impairs antitumorigenic type I interferon responses in hepatocellular carcinoma. *Immunity* **56**, 180-192 (2023).
13. Lu, L.G. *et al.* PD-L1 blockade liberates intrinsic antitumorigenic properties of glycolytic macrophages in hepatocellular carcinoma. *Gut* **71**, 2551-2560 (2022).
14. Li, Q. *et al.* PRDM1/BLIMP1 induces cancer immune evasion by modulating the USP22-SPI1-PD-L1 axis in hepatocellular carcinoma cells. *Nat Commun* **13**, 7677 (2022).
15. Bai, A.P. *et al.* NADH oxidase-dependent CD39 expression by CD8⁺ T cells modulates interferon gamma responses via generation of adenosine. *Nat Commun* **11**, 3036 (2020).
16. Chow, A. *et al.* Tim-4 cavity-resident macrophages impair anti-tumor CD8⁺ T cell immunity. *Cancer Cell* **39**, 973-988 (2021).
17. Gu, X.M. *et al.* Itaconate promotes hepatocellular carcinoma progression by epigenetic induction of CD8⁺ T-cell exhaustion. *Nat Commun* **14**, 8154 (2023).

REVIEWERS' COMMENTS

Reviewer #1 (Remarks to the Author):

None, they addressed my concerns

Reviewer #2 (Remarks to the Author):

The authors addressed all my questions and improved the quality of data and findings' reliability significantly.

Reviewer #3 (Remarks to the Author):

The authors did a great job and addressed most of my concerns. The revised manuscript has been improved significantly, and publication is suggested.

Point-by-point response to the comments

Reviewer #1 (Remarks to the Author).

Comments 1-1:

None, they addressed my concerns.

Response: We are grateful for the positive comments provided by our reviewer.

Reviewer #2 (Remarks to the Author):

Comments 2-1:

The authors addressed all my questions and improved the quality of data and findings' reliability significantly.

Response: We appreciate our reviewer for positive comments.

Reviewer #3 (Remarks to the Author):

Comments 3-1:

The authors did a great job and addressed most of my concerns. The revised manuscript has been improved significantly, and publication is suggested.

Response: We thank the reviewer for the positive and encouraging comments.